# Du-IN: Discrete units-guided mask modeling for decoding speech from Intracranial Neural signals

**Hui Zheng**[*,2,4], **Hai-Teng Wang**[*,1], **Wei-Bang Jiang**[3], **Zhong-Tao Chen**[1],
**Li He**[1], **Pei-Yang Lin**[1], **Peng-Hu Wei**[5], **Guo-Guang Zhao**[5], **Yun-Zhe Liu**[†,1,4]

[1]Beijing Normal University, [2]Peking University, [3]Shanghai Jiao Tong University,
[4]Chinese Institute for Brain Research, [5]Capital Medical University, Xuanwu Hospital, Beijing
*Equal contribution,†yunzhe.liu@bnu.edu.cn

## Abstract

Invasive brain-computer interfaces with Electrocorticography (ECoG) have shown promise for high-performance speech decoding in medical applications, but less damaging methods like intracranial stereo-electroencephalography (sEEG) remain underexplored. With rapid advances in representation learning, leveraging abundant recordings to enhance speech decoding is increasingly attractive. However, popular methods often pre-train temporal models based on brain-level tokens, overlooking that brain activities in different regions are highly desynchronized during tasks. Alternatively, they pre-train spatial-temporal models based on channel-level tokens but fail to evaluate them on challenging tasks like speech decoding, which requires intricate processing in specific language-related areas. To address this issue, we collected a well-annotated Chinese word-reading sEEG dataset targeting language-related brain networks from 12 subjects. Using this benchmark, we developed the Du-IN[1] model, which extracts contextual embeddings based on region-level tokens through discrete codex-guided mask modeling. Our model achieves state-of-the-art performance on the 61-word classification task, surpassing all baselines. Model comparisons and ablation studies reveal that our design choices, including (i) temporal modeling based on region-level tokens by utilizing 1D depthwise convolution to fuse channels in the ventral sensorimotor cortex (vSMC) and superior temporal gyrus (STG) and (ii) self-supervision through discrete codex-guided mask modeling, significantly contribute to this performance. Overall, our approach – inspired by neuroscience findings and capitalizing on region-level representations from specific brain regions – is suitable for invasive brain modeling and represents a promising neuro-inspired AI approach in brain-computer interfaces. Code and dataset are available at https://github.com/liulab-repository/Du-IN.

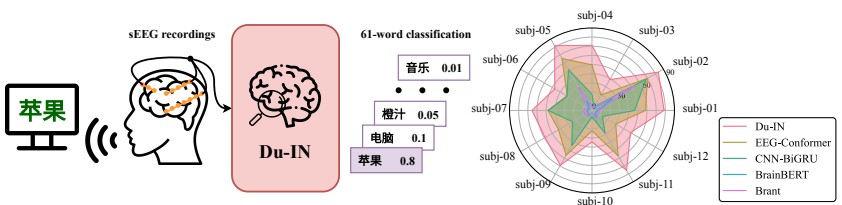

Figure 1: Overall illustration of sEEG decoding setup and comparison with SOTA baselines.

---

[1]Du-IN refers to the phonetic transcription of "讀音" (i.e., pronunciation) in Chinese.

38th Conference on Neural Information Processing Systems (NeurIPS 2024).

# 1 Introduction

Brain signals refer to the biometric information collected from the brain. Their patterns provide valuable insights toward understanding the physiological functions of the brain and the mechanism of related diseases, leading to various applications, including speech decoding [13, 19, 37], sleep cognition research [35, 55], neurological disorders detection [28, 54], and so on. Due to the high signal-noise ratio, invasive recording methods (e.g., stereoElectroEncephaloGraphy (sEEG), ElectroCorticoGraphy (ECoG)) usually reveal these underlying mechanisms better than non-invasive recording methods. Many previous works [29, 19] have shown that decoding speech from EEG signals is difficult, and the performance is limited. Compared with ECoG, sEEG imposes less trauma on patients and provides more stereotactic information from specific brain regions. Although some studies [37, 36] have recently shown promise for building high-performance speech decoders based on ECoG, there are few attempts made to explore the potential of sEEG-based speech decoding.

Modeling intracranial neural signals, especially sEEG, has gained significant attention, but several issues remain unresolved. Current research on modeling neural signals is divided into two lines based on the basic modeling units (e.g., channel-level tokens or group-level tokens[2]). Some studies [54, 28] utilize shared embedding blocks to embed single channels into channel-level tokens, neglecting the specificity of brain computation [8]; then they adopt spatial-temporal integration to model spatial relationships among them, attempting to regain the precise state of the brain. However, these methods mainly focus on channel-level classification tasks, e.g., seizure detection, yet fail to validate them on more challenging group-level classification tasks, e.g., speech decoding. Other studies [19, 21] fuse all channels (across the brain) to build brain-level tokens, overlooking the brain's desynchronization nature [7]; then they adopt temporal modeling to capture the rapid process of brain dynamics. Besides, labeling data at scale in medical experiments is often impractical or costly, emphasizing the need to maximize label efficiency. Hence, developing an efficient pre-training framework that draws on prior neuroscience findings is highly appealing, as it can make the most of abundant unlabeled data.

The primary challenge in modeling intracranial neural signals lies in extracting meaningful tokens, requiring careful consideration of two key factors. (1) Temporal scale. Since intracranial neural signals have high temporal resolution and signal-noise ratio, these tokens must capture rapid dynamic changes in brain activity. (2) Spatial scale. Considering the brain's desynchronization nature, these tokens should correctly capture the information of each brain region for further integration and, if needed, decouple different parts of brain dynamics within each brain region. To better assess how well different models capture the intricate processing within each brain region, we can evaluate these methods on tasks mainly involving a few brain regions.

Since speech mainly involves specific brain regions related to vocal production, as demonstrated in Section 2.1, we utilize speech decoding tasks to evaluate which model can effectively extract information from specific brain regions. Since there are too few open source sEEG language datasets [1, 49], we collected a well-annotated Chinese word-reading sEEG dataset (vocal production), including 12 subjects, which makes up for the problem of missing sEEG recordings in language tasks. Inspired by neuroscientific findings, we systematically demonstrate the locality and specificity of brain computation and propose the Du-IN model to solve the abovementioned issues. Compared to other existing methods for modeling brain signals, Du-IN achieves SOTA performance on the 61-word classification task, demonstrating the effectiveness of our model in extracting meaningful tokens that can capture both the rapid changes and the precise state of specific brain regions. It marks a promising neuro-inspired AI approach [42, 41] in BCI.

To sum up, the main contributions of our work comprise:

1. A well-annotated Chinese word-reading sEEG dataset, addressing the lack of sEEG language dataset. The dataset will be publicly available.

2. Demonstration of brain-specific computation – achieving the best decoding performance only requires about one electrode in specific brain regions (i.e., vSMC, STG).

3. A novel framework for sEEG speech decoding – Du-IN, which learns region-level contextual embeddings through discrete codex-guided mask modeling.

4. SOTA performance on the sEEG speech decoding task – Du-IN achieves 62.70% top-1 accuracy on the 61-word classification task, surpassing all other baselines.

---

[2]The term "group-level" includes "brain-level" and "region-level," and is distinct from "channel-level."

## 2 Related Works

### 2.1 Neural Basis of Language Function

Neuroscientific research [5, 17, 43] in the past has extensively explored brain regions supporting language functionality. In neuroscience, the investigation into language functionality related to speech has been categorized into two main streams: one dedicated to semantic processing and the other to vocal production. Previous studies [4, 43] have shown that brain regions associated with semantic processing primarily include left inferior frontal gyrus (IFG), left anterior temporal lobe (ATL), and bilateral middle temporal gyrus (MTG).

As for vocal production, which is also the focus of our work, it is predominantly governed by motor information related to language articulation, primarily involving ventral sensorimotor cortex (vSMC), bilateral superior temporal gyrus (STG), and bilateral dorsal laryngeal motor cortex (dLMC) [5, 17, 9]. Our analysis results based on our collected word-reading sEEG dataset also confirm this point, as illustrated in Figure 4.

### 2.2 Language Decoding in BCI

The keys to decoding natural language from brain signals are (1) high-quality recordings, and (2) well-designed models with good representations. Compared to non-invasive recordings (e.g., EEG), invasive recordings manifest advantages in providing detailed information about specific brain regions with a high signal-noise ratio. Since speech mainly involves some specific brain regions, obtaining detailed recordings of these brain regions will significantly enhance the decoding performance. Existing works [13, 37, 21] have shown the great potential of building a high-performance decoder based on invasive recordings.

The other key is well-designed models with good representations. Existing work for brain-to-language representations can be classified into two categories: self-supervision or alignment with representation models pre-trained on other modalities (e.g., text, audio). BrainBERT [49] learns general embeddings through self-supervised mask modeling. DeWave [19] introduces discrete codex encoding and aligns neural representations with text embeddings from BART [32], thus enhancing the extraction of semantic processing-related information from EEG recordings. Metzger et al. [36] align neural representations with acoustic embeddings to improve the extraction of vocal production-related information from ECoG recordings.

### 2.3 Self-supervised Learning in BCI

In recent years, self-supervised pre-training has made significant progress in natural language processing [16, 39, 6] and computer vision [3, 25, 11]. However, its potential in BCI is far from being explored. BrainBERT (for sEEG) [49] embeds single channels into channel-level tokens and utilizes mask modeling to learn general representations. Brant (for sEEG) [54, 53], PopT (for sEEG) [10] and some works (for EEG) [28, 23] further adopt spatial-temporal integration to model spatial relationships among them. Some works (for EEG) [31, 20, 50, 22] take the other way – fusing all channels (across the whole brain) to build brain-level tokens, and it uses self-supervised learning to learn contextual representations. Considering the difference among brain regions, MMM (for EEG) [52] further splits channels into different groups to build region-level tokens.

All existing pre-training methods for sEEG primarily pre-train spatial-temporal models based on channel-level tokens yet only evaluate them on channel-level classification tasks, e.g., seizure detection. However, unlike EEG pre-training methods, their effectiveness over more challenging group-level classification tasks, e.g., speech decoding. Besides, there is no standard channel configuration for sEEG recordings, unlike EEG recordings, which makes modeling spatial relationships in sEEG more challenging.

## 3 Method

The overall architecture of Du-IN is illustrated in Figure 2, where the raw sEEG signals are fused across channels to build region-level tokens and further encoded for downstream tasks.

## 3.1 Task Definition

Due to the lack of open-source sEEG datasets related to language tasks, we follow the experimental design outlined by Moses et al. [37] to collect a well-annotated Chinese word-reading sEEG dataset (vocal production). During the experiment, each subject speaks aloud 61 pre-determined Chinese words 50 times; see Appendix A for more details. We formulate the multi-channel sEEG signals as $\mathcal{X} \in \mathbb{R}^{C \times T}$, where $C$ is the number of sEEG channels and $T$ is the total timestamps. The associated word label is denoted as $y \in \mathcal{Y}$, where $\mathcal{Y}$ represents the set of 61 pre-determined words. In summary, this dataset comprises paired sEEG-word data ($\langle \mathcal{X}, y \rangle$), and the model aims to decode the corresponding word $y$ from a sequence of raw sEEG signals $\mathcal{X}$.

## 3.2 Model Architecture

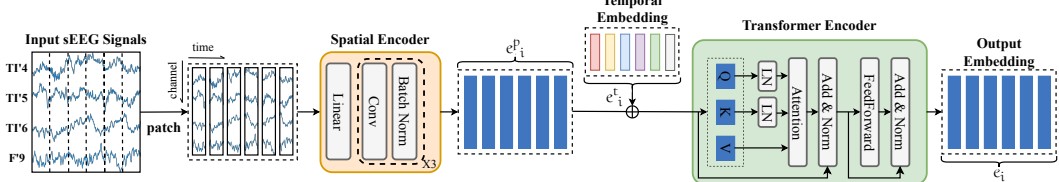

Figure 2: **The overall architecture of Du-IN Encoder.** Du-IN Encoder is used as an encoder in all Du-IN models (i.e., Du-IN VQ-VAE, Du-IN MAE, Du-IN CLS (classification)), see Appendix C for more details.

We introduce the Du-IN Encoder, a general architecture for sEEG speech decoding tasks that can deal with any input sEEG signals with arbitrary time length, as shown in Figure 2. The key operation for archiving this is segmenting the sEEG signals into patches, inspired by patch embeddings in images [18]. For each sample $\mathcal{X}$, we use a $W$-length window without overlap to segment it into patches, obtaining $\mathcal{X} = \{\boldsymbol{x}_i \in \mathbb{R}^{C \times W} | i = 1, ..., N\}$, where $N = \lfloor \frac{T}{W} \rfloor$ is the number of patches.

**Spatial Encoder.** As each sEEG patch has multiple channels, it is vital to fuse different channels to extract meaningful features before patch-wise interaction by self-attention. We employ a spatial encoder, which consists of a linear projection and several convolution blocks, to encode each sEEG patch into a patch embedding. The linear projection transforms the raw sEEG signals into the hidden neural space, and its weights are utilized for subsequent analysis. The convolution block is composed of a 1D depthwise convolution layer and a batch normalization layer [27]. We denote the output patch embeddings from the spatial encoder as

$$\mathcal{E}_p = \{\boldsymbol{e}_i^p \in \mathbb{R}^d | i = 1, ..., N\}, \tag{1}$$

where $d$ is the dimension of the embeddings.

**Temporal Embedding.** In order to enable the model to be aware of the temporal information of patch embeddings, we utilize the parameter-free position embeddings introduced in [48], i.e., $\mathcal{E}_t = \{\boldsymbol{e}_1^t, ..., \boldsymbol{e}_{t_{max}}^t\}$. Note that $t_{max}$ is the hyperparameter determining the maximum number of time patches and $t_{max} \geq N$. Given one arbitrary patch embedding $\boldsymbol{e}_i$ in Equation 1 from the spatial encoder, we add the corresponding temporal embedding to it:

$$\mathcal{E}_{init} = \{\boldsymbol{e}_i^p + \boldsymbol{e}_i^t | i = 1, ..., N\}, \tag{2}$$

which forms the input embeddings $\mathcal{E}_{init}$ for the Transformer Encoder.

**Transformer Encoder.** Finally, the sequence of embeddings will be directly fed into the Transformer encoder [48] to get the final encoded $\mathcal{E} = \{\boldsymbol{e}_i \in \mathbb{R}^d | i = 1, ..., N\}$. To make the training of the Transformer more stable and efficient, we incorporate some modifications [14] inspired by LaBraM [28]. We add layer normalization to the queries and keys before the dot-product attention mechanism, which avoids over-large values in attention logits:

$$\text{Attention}(Q, K, V) = \text{softmax}\left(\frac{\text{LN}(Q)\text{LN}(K)^T}{\sqrt{d_{head}}}\right)V, \tag{3}$$

where $d_{head}$ is the dimension of attention head and LN denotes layer normalization [2]. For downstream classification tasks, we flatten the output embeddings followed by a classification head.

### 3.3 Du-IN VQ-VAE Training

Prior to pre-training Du-IN through mask modeling, we need to tokenize the sEEG patches into discrete tokens. We introduce vector-quantized neural signal regression, which is trained by reconstructing the original sEEG signals, as shown in Figure 3. The key components are the Du-IN Encoder, which encodes the raw sEEG samples into embeddings, and the Du-IN Regressor, which reconstructs the original sEEG signals. The idea is basically inspired by VQ-VAE [47], which encodes images into discrete latent embeddings.

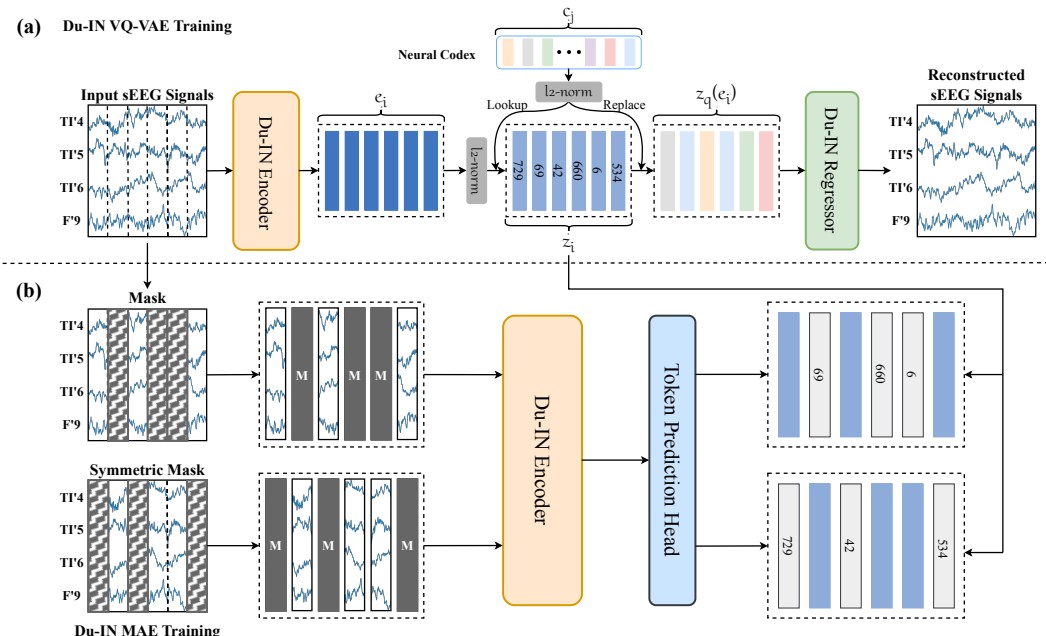

Figure 3: **Overview of Du-IN VQ-VAE training and Du-IN MAE training. (a).** We train the Du-IN Encoder in the Du-IN VQ-VAE to discretize sEEG signals into discrete neural tokens by reconstructing the original sEEG signals. **(b).** During the training of Du-IN MAE, part of sEEG patches are masked while the objective is to predict masked tokens from visible patches.

**Du-IN Encoder.** We define a neural codex $\mathcal{C} = \{c_j | j = 1, ..., N_{codex}\} \in \mathbb{R}^{N_{codex} \times d_{codex}}$, where $N_{codex}$ is the number of the discrete neural embeddings and $d_{codex}$ is the dimension of each embedding. Given a sEEG sample $\mathcal{X}$, the Du-IN Encoder, illustrated in Figure 2, first encodes it to embeddings $\mathcal{E} = \{e_i \in \mathbb{R}^d | i = 1, ..., N\}$. After that, we utilize a linear projection $\mathbf{z}_c$ to get the mapped embeddings $\mathbf{z}_c(\mathcal{E}) = \{\mathbf{z}_c(e_i) \in \mathbb{R}^{d_{codex}} | i = 1, ..., N\}$ in the codex space. Then, the codex looks up the nearest neighbor of each embedding $\mathbf{z}_c(e_i)$ in the neural codex $\mathcal{C}$. This procedure can be formulated as

$$\mathbf{z}_q(\mathcal{E}) = \{\mathbf{z}_q(e_i) | i = 1, ..., N\}, \quad \mathbf{z}_q(e_i) = c_{z_i}, \quad z_i = \arg \min_j ||\ell_2(\mathbf{z}_c(e_i)) - \ell_2(c_j)||_2, \quad (4)$$

where $\ell_2$ represents $\ell_2$ normalization and $\mathbf{z}_q(e_i)$ is the quantized vector after the quantizer. This is equivalent to finding the closest neural embedding by cosine similarity and such $\ell_2$ normalization improves the codex utilization [38].

**Du-IN Regressor.** The Du-IN Regressor consists of a Transformer decoder and a stack of transposed convolution layers. Given a sequence of the vector-quantized embeddings $\mathcal{Z} = \{z_i | i = 1, ..., N\}$, the Du-IN Regressor convert these discrete embeddings back into raw sEEG signals $\tilde{\mathcal{X}} = \{\tilde{x}_i | i = 1, ..., N\}$. The mean squared error (MSE) loss is utilized to guide the regression. The total loss for training the Du-IN VQ-VAE is defined as:

$$\mathcal{L}_{vqvae} = \sum_{i=1}^{N} \left[ ||\tilde{x}_i - x_i||_2^2 + ||\mathbf{sg}[\mathbf{z}_c(e_i)] - \mathbf{z}_q(e_i)||_2^2 + \beta ||\mathbf{z}_c(e_i) - \mathbf{sg}[\mathbf{z}_q(e_i)]||_2^2 \right], \quad (5)$$

where **sg** represents the stop-gradient operation, which is an identity at the forward pass and has zero gradients. To stabilize the codex update, we use the exponential moving average strategy [47].

### 3.4 Pre-training Du-IN

**Masked sEEG Modeling.** To enforce Du-IN learning contextual representations, we propose masked sEEG modeling. The whole procedure is presented in Figure 3. As illustrated in Figure 2, given a sEEG sample $\mathcal{X}$, the spatial encoder first transforms it to patch embeddings $\mathcal{E}_p = \{e_i^p | i = 1, ..., N\}$. Given these patch embeddings $\mathcal{E}_p$, around 50% of patch embeddings are patch-wisely chosen and masked. The masked position is termed as $\mathcal{M}$. Then, a shared learnable embedding $e_{[M]} \in \mathbb{R}^d$ is used to replace the original patch embeddings:

$$\mathcal{E}_m = \{e_i^m | i = 1, ..., N\}, \quad e_i^m = m_i \odot e_{[M]} + (1 - m_i) \odot e_i^p, \tag{6}$$

where $\delta(\cdot)$ is the indicator function and $m_i = \delta(i \in \mathcal{M})$. After that, the masked embeddings $\mathcal{E}_m$ will be added by temporal embeddings, and then fed into the Transformer encoder. The output embeddings $\mathcal{E}$ will be used to predict the indices of the corresponding codes from the codex in the Du-IN VQ-VAE through a linear classifier:

$$p(z_i | e_i) = \text{softmax}(\text{Linear}(e_i)), \tag{7}$$

The training loss of mask modeling is defined as:

$$\mathcal{L}_{\mathcal{M}} = -\sum_{i \in \mathcal{M}} m_i \odot \log p(z_i | e_i). \tag{8}$$

**Symmetric Masking.** Inspired by LaBraM [28], we further introduce a symmetric masking strategy to improve training efficiency. We calculate the inverse of the generated mask $\mathcal{M}$, obtaining $\hat{\mathcal{M}}$. Similarly, we use the new mask $\hat{\mathcal{M}}$ to perform the mask modeling, obtaining the mask modeling loss $\mathcal{L}_{\mathcal{M}}^{sym}$. The total loss for pre-training the Du-IN model (i.e., Du-IN MAE model) is defined as:

$$\mathcal{L}_{mae} = \mathcal{L}_{\mathcal{M}} + \mathcal{L}_{\mathcal{M}}^{sym}. \tag{9}$$

## 4 Experiments

### 4.1 Dataset

Due to the lack of open-source sEEG datasets related to language tasks, we follow the experimental design outlined by Moses et al. [37] to collect a well-annotated Chinese word-reading sEEG dataset (vocal production), including 12 subjects. The subjects undergo a surgical procedure to implant 7 to 13 invasive sEEG electrodes, each with 72 to 158 channels, in their brain. For each subject, the dataset contains 15 hours of 2000Hz recordings, 3 hours of which are task recordings.

**Pre-training dataset.** For each subject, the pre-training dataset contains all sEEG recordings (with about 54 million timestamps) of that subject. To stabilize computing resource usage, the time length of sEEG sample $\mathcal{X}$ is set to 4 seconds.

**Downstream dataset.** For each subject, 3 hours of the sEEG recordings are task recordings. The sEEG signals are segmented into about 3000 3-second samples, each of which is paired with the corresponding word label (from 61 pre-determined words).

### 4.2 Implementation Details

**Preprocess.** We first filter the sEEG signals between 0.5Hz and 200Hz to remove low-frequency noise. Then, a notch filter of 50Hz is applied to avoid power-line interference. After that, all sEEG signals are resampled to 1000Hz and bi-polar re-referenced [33]. Finally, we perform z-score normalization on each channel to guarantee normalized data scales across all channels.

**Model Configurations.** The length of the sEEG patch is 100ms, resulting in 40 patches per sample in the pre-training dataset and 30 patches per sample in the downstream dataset. The "Spatial Encoder" contains one linear projection and three 1D convolution layers, transforming the original sEEG patches into patch embeddings with $d = 160$. The following "Transformer Encoder" contains an 8-layer Transformer encoder with model dimension $d = 160$, inner dimension (FFN) $d_{ff} = 320$, and 8 attention heads. See Appendix C for more details.

**Pre-training.** During the pre-training, we use either all sEEG recordings (15 hours) or the sEEG recordings without task recordings (12 hours) to train the Du-IN VQ-VAE and Du-IN MAE models. To enhance the robustness of the learned codex and representations, we further use data augmentation described in Appendix D. For each subject, the model is pre-trained on a Linux system with 2 CPUs (Intel Xeon Gold 6230 40-Core Processor) and 1 GPU (NVIDIA Tesla V100 32GB) for $\sim 1.2$ days.

**Fine-tuning.** During the downstream evaluation, we split the task recordings into training, validation, and testing splits with a size roughly proportional to 80%, 10%, and 10%. All experiments are conducted on the same machine with the same set of random seeds. The train/validation/test splits are the same across different models. We also use data augmentation, as described in Appendix D, to make the most of the gathered dataset. We employ cross-entropy loss (multi-class classification) as the training loss. Our experiments are conducted on one V100 GPU by Python 3.11.7 and PyTorch 2.1.2 + CUDA 12.3. The best models are trained based on the training set, selected from the validation set according to accuracy, and finally evaluated on the test set. For model comparison, we report the average and standard error values (of all subjects) on six different random seeds to obtain comparable results. For the results of the subject-wise evaluation, we report the average and standard deviation values (of each subject) in Appendix K.

## 4.3 Channel Contribution and Selection

As demonstrated in Section 2.1, previous neuroscience studies reveal that vocal production predominantly engages specific brain regions. Given the sparse distribution of implanted sEEG electrodes (each containing 8-16 channels), it's vital to exclude redundant electrodes unrelated to vocal production, thus improving decoding performance. We retain electrodes implanted in relevant brain regions and evaluate the performance based on the remaining electrodes. Table 1 demonstrates that excluding approximately 85% electrodes even leads to a dramatic increase in decoding performance.

Table 1: The performance of Du-IN with or without electrode selection.

| Methods | # of Channels (Averaged) | Accuracy (%) $\pm$ Ste (%) |
|---|---|---|
| Du-IN (w/o electrode selection) | 109.75 | 30.12$\pm$5.64 |
| Du-IN (w/ electrode selection) | 12.25 | 55.92$\pm$4.96 |

To further understand the detailed contribution of each channel, we analyze the weights of linear projection in the spatial encoder. In detail, we calculate the contribution scores of channels per subject and organize them accordingly, as described in Appendix H. Figure 4 demonstrates that (1) the brain regions effective for speech decoding align with findings from previous neuroscience research, and (2) our model achieves optimal decoding performance with approximately 10 channels, 80% of which originate from the same electrode. To streamline, we utilize these top 10 channels (selected according to train-set) for both pre-training and downstream evaluation.

## 4.4 Comparasion with Other Models

Table 2 presents the results of our Du-IN model and the advanced baselines that are designed for either brain signals or general time series. See Appendix B and Appendix C.3 for detailed descriptions of models. The results demonstrate that our Du-IN model outperforms all baselines. It's worth noting that the models (i.e., the foundation models designed for brain signals) that adopt spatial-temporal integration to model spatial relationships among channel-level tokens perform worse than the models that adopt temporal modeling based on region-level tokens, challenging the generalizability of current strategies to model spatial relationships among channels with Transformer.

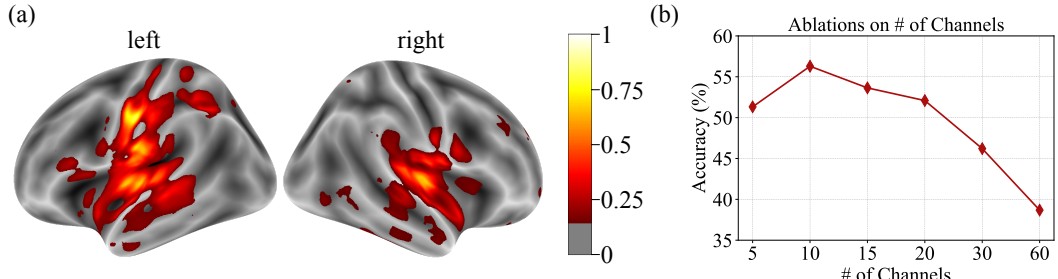

Figure 4: **The channel contribution analysis. (a).** The channel contribution map. **(b).** The effect of the number of channels (sorted according to channel contribution scores) on decoding performance.

Table 2: The performance of different methods (with the best in **bold** and the second underlined).

| Methods | Token Level | Config PT[1] | Config MS[2] | Model Size | Accuracy (%) $\pm$ Ste (%) |
|---------|-------------|-----|-----|------------|---------------------------|
| TS-TCC[20] | Region | ✓ | ✗ | 0.32M | 24.85±4.42 |
| CNN-BiGRU[37] | Region | ✗ | - | 0.54M | 32.04±5.45 |
| EEG-Conformer[44] | Region | ✗ | - | 2.34M | 45.82±4.66 |
| Neuro-BERT[50] | Region | ✓ | ✗ | 2.14M | 49.51±4.43 |
| DeWave[19] | Region | ✗ | - | 5.70M | 32.43±4.48 |
| BrainBERT[49] | Channel | ✓ | ✗ | 43.58M | 6.72±1.59 |
| BrainBERT[49] | Channel | ✓ | ✓ | 43.58M | 7.50±1.76 |
| Brant[54] | Channel | ✓ | ✗ | 69.35M | 11.16±3.56 |
| Brant[54] | Channel | ✓ | ✓ | 69.35M | 12.42±4.10 |
| LaBraM[28] | Channel | ✓ | ✗ | 6.85M | 11.53±2.63 |
| LaBraM-PopT[28, 10] | Channel | ✓ | ✓ | 6.85M | 11.78±2.70 |
| Du-IN | Region | ✗ | - | 4.38M | 56.29±5.20 |
| Du-IN (vqvae+vq) | Region | ✓ | ✗ | 4.38M | 44.17±4.04 |
| Du-IN (vqvae) | Region | ✓ | ✗ | 4.38M | 58.24±4.83 |
| Du-IN (mae) | Region | ✓ | ✗ | 4.38M | **62.70±4.69** |
| Du-IN (poms) | Region | ✓ | ✓ | 5.18M | 59.18±4.63 |

[1] PT: Whether the model is pre-trained before evaluation.
[2] MS: Whether the model is pre-trained across multiple subjects.

As BrainBERT [49] doesn't consider the spatial relationships among channels, we mainly focus on understanding why Brant [54], LaBraM [28] and LaBraM-Popt [28, 10] fail to effectively capture the discriminative features on the speech decoding task. These models typically build channel-level tokens by segmenting non-overlapping patches with large receptive fields (e.g., 1 second) from single channels. However, this approach makes it challenging to capture the rapid process of brain dynamics. Moreover, while these models further utilize Transformer to capture the spatial relationships among these tokens, they do not encourage region-level embeddings, either through their architecture [52] or their pre-training objective [10]. Therefore, the effectiveness of building brain foundation models based on these spatial-temporal backbones is still under exploration, especially for cognitive tasks (e.g., speech decoding), which are of great value in the field of neuroscience.

Besides, unlike LaBraM [28], Brant doesn't introduce spatial embeddings to identify the spatial location of each channel. Since the electrodes are sparsely distributed in the brain and the raw sEEG signals on the same electrode are highly correlated, it's fairly easy to identify their spatial relationships through their values. As demonstrated in iTransformer [34], this modeling approach is well suited for detecting time-delay events, e.g., seizure detection. For speech decoding tasks,

sEEG often requires bi-polar re-reference (or Laplacian re-reference) to remove the high correlations among channels, thus avoiding model overfitting [49]. Once the correlations among channels have been removed, Brant will lose the ability to model spatial relationships among channels.

For other baselines that use temporal modeling based on region-level tokens, we provide a detailed explanation of their performance differences as follows. TS-TCC [20] tokenizes raw sEEG signals into region-level tokens with a stack of 1D depthwise convolution blocks, but it lacks a temporal Transformer for further integration over time. CNN-BiGRU [37] introduces a stack of GRU layers on top of these tokens to perform temporal integration. EEG-Conformer [44] introduces a temporal Transformer to better integrate global temporal information, which makes it outperform CNN-BiGRU. However, EEG-Conformer tokenizes raw sEEG signals with the temporal-spatial convolution, applying the same convolutional kernel across different channels, which overlooks the specificity of brain computation [8]. This also raises a challenge for the effectiveness of current sEEG foundation models, which rely on shared convolution blocks across individual channels. Neuro-BERT [50] further introduces mask modeling to learn contextual embeddings, which makes it outperform EEG-Conformer. DeWave [19] utilizes the Conformer model [24] for tokenization, which involves more parameters but is less effective than 1D depthwise convolution.

### 4.5 Ablation Study

**Self-Supervision Initialization.** As illustrated in Figure 3, the Du-IN model entails a two-stage pre-training process, wherein both the Du-IN VQ-VAE model and the Du-IN MAE model are trained. Previous studies utilize different strategies [19, 12, 28] to leverage these pre-trained models to enhance the performance of downstream tasks. Here, we evaluate these different strategies for comparison; see Appendix C.3 for detailed definitions. Table 2 shows that initializing weights from the Du-IN MAE model captures contextual embeddings effectively, resulting in the highest decoding performance.

**Pre-training with/without Downstream Datasets.** During the pre-training stage, we hope that the Du-IN VQ-VAE model can extract general tokens of that brain region, thus guiding the Du-IN MAE model to learn general representations that are not specific to any particular task. Although no label data is used during the pre-training stage, to eliminate the influence of the pre-training data on downstream tasks, we compare the results with or without incorporating the downstream task dataset into the pre-training stage. Table 3 shows a slight performance drop when excluding downstream datasets. However, the decoding performance is still higher than the baseline performance without pre-training, which means that the degradation is mainly due to the decrease of the pre-training dataset. We hope that, with more pure recordings, our model can achieve better decoding performance.

Table 3: Ablation study on whether pre-training with the downstream dataset (DD) or not.

| Methods | Pre-training Dataset Size | Accuracy (%) $\pm$ Ste (%) |
|---|---|---|
| Du-IN (mae w/o DD) | 12 hours per subject | 60.02$\pm$4.34 |
| Du-IN (mae w/ DD) | 15 hours per subject | 62.70$\pm$4.69 |

**Discrete Codex.** During the Du-IN VQ-VAE training stage, the Du-IN VQ-VAE model encodes sEEG patches into discrete codes and then reconstructs the original signal from these codes. We evaluate performance against varying codex sizes (512 to 8192) to ascertain if codex size affects the quality of the learned codex. As illustrated in Figure 5, while extremely small codex size lacks representation diversity, extremely large codex size often leads to codex collapse. We suspect that our existing training data might not be adequate for larger codex sizes. Furthermore, our experiments suggest that the model performs optimally when the codex dimension, denoted as $d_{codex} = 64$, is slightly less than the model dimension, $d = 160$, yielding a more effective regularization effect.

**Perception Time Window.** We also conduct the ablation study on the model structure for the spatial encoder described in Section 3.2. As the spatial encoder transforms the sEEG signals within a given patch to a patch embedding, it compresses the sEEG signals for perception. As described in Section 4.2, the model utilizes a receptive field of 100ms. We conduct an ablation study of different receptive fields and report it in Figure 5. The model performance notably drops with a receptive field smaller than 60ms and gradually declines as the receptive field exceeds 160ms. The model reaches a

small peak around 100ms to 140ms. We think this phenomenon is rational since sEEG is known for its ability to capture the rapid dynamics of specific brain regions precisely.

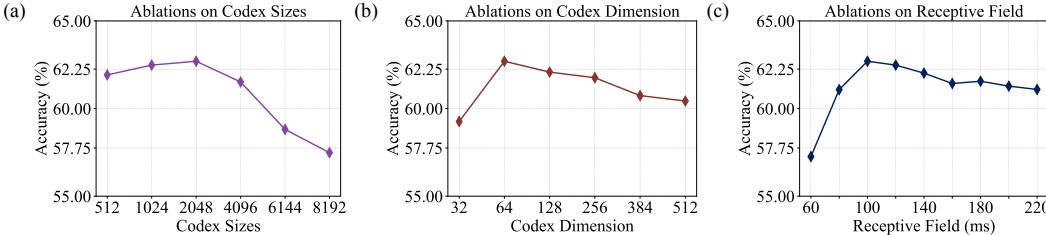

Figure 5: Ablation study on different codex sizes, codex dimensions, and receptive fields.

## 5 Limitations

Despite Du-IN's enhancements in speech decoding via discrete codex-guided mask modeling, it is still restricted to close-set speech decoding tasks (i.e., the word set only includes 61 pre-determined words). However, a parallel to our work, Feng et al. [21], which follows previous works [26, 45], build an acoustic-inspired model that can decode arbitrary Chinese words by predicting syllable components (initials, finals, tones). Although their method requires a large amount of labeled data, their experimental design mirrors ours closely. The difference lies in the requirement for the subject to repeat syllable components, instead of entire words. Therefore, with slight modifications, our model can support open-set speech decoding tasks.

Additionally, the experiments in this paper are restricted to the vocal production part of language decoding, i.e., speech decoding. A more interesting but difficult task is to decode language from the semantic level, in which large language models have been wildly used to improve the model performance [46, 19]. However, due to the locality of sEEG recordings, it is still under exploration whether sEEG recordings can fully capture semantic-related information across brain regions.

## 6 Conclusion

This paper proposes Du-IN, a framework for speech decoding, which learns contextual embeddings through discrete codex-guided mask modeling on specific brain regions. To evaluate our model, we collect a well-annotated Chinese word-reading sEEG dataset to address the lack of sEEG language dataset. Inspired by neuroscientific findings, we analyze the effective brain regions for speech decoding and achieve the best decoding performance with about one electrode in specific brain regions, which dovetails with the past neuroscientific research on language. Comprehensive experiments demonstrate that our model outperforms both supervised and sEEG-based self-supervised baselines, effectively capturing the intricate processing within specific brain regions. It marks a promising neuro-inspired AI approach in BCI. In the end, we hope our work can have implications for future developments in sEEG-based self-supervised models with more consideration over how to build the basic representation units so that the model can maximally benefit from the pre-training stage.

## 7 Broader Impacts

Our method advances the feasibility of invasive BCI technology by being the first to demonstrate speech decoding using a single sEEG electrode, which holds significant potential for clinical applications. For patients who have lost their ability to communicate or perform daily tasks due to neurological conditions like locked-in syndrome or amyotrophic lateral sclerosis (ALS), our approach offers a less invasive alternative to technologies like ECoG or microelectrode arrays, thereby reducing the risk of brain damage.

**Acknowledgements**

This study was supported by the National Science and Technology Innovation 2030 Major Program (2022ZD0205500), the National Natural Science Foundation of China (32271093), the Beijing Natural Science Foundation (Z230010, L222033), and the Fundamental Research Funds for the Central Universities. We would like to extend our sincere appreciation to Dr. Zhi-Feng Yue for his coordination and support in securing the computing resources essential for this study. Besides, we also sincerely appreciate the LaBram [28] team for their valuable discussion on the visual design of Figure 2 and Figure 3.

**Ethics Statement**

Experiments that contribute to this work were approved by IRB. All subjects consent to participate. All electrode locations are exclusively dictated by clinical considerations.

Our informed consent signing process is as follows:

1. If the experimental participants are adults and have full civil capacity, we will ask them to sign a written informed consent after the participants have fully informed consent;

2. If the experimental participants are minors or do not have full civil capacity, we will ask the participant's legal guardian to sign a written informed consent after the participants and their legal guardians have fully informed consent.

Our informed consent form includes the following points:

1. Contact information of research institutions and researchers;

2. Research direction and purpose;

3. Risks involved in the research;

4. Personal information, data and usage methods to be used in the research;

5. Privacy protection statement (all personal identification information (PII) will not be disclosed);

6. Data storage statement (retained after deleting all personal identification information (PII));

7. Voluntary statement of participants;

8. Statement that participants can withdraw unconditionally at any time.

Our data storage and protection procedures include the following processes:

1. Our data collection, transfer, and analysis tasks are only completed by researchers who have signed relevant confidentiality agreements;

2. The collected raw data will be copied twice as soon as possible, one copy to a storage computer that is not connected to the Internet and encrypted, and the other copy to a mobile hard disk and encrypted and stored offline;

3. The use of the data is only authorized to the research leader and the main researchers (less than 5 people), among which the main researchers can only access data that does not contain personal identification information (PII);

4. After the study is completed, all personal identification information (PII) on both nodes (storage computer, mobile hard disk) will be deleted immediately.

To prevent unauthorized access or possible data leakage, we use double encryption on the storage computer, that is, a static password and a dynamic password (received by mobile phone or email); physical isolation is used on the mobile hard disk, that is, it is locked in a filing cabinet, and the key is only kept by the research leader and the main researchers.

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

# A  Experiment Design

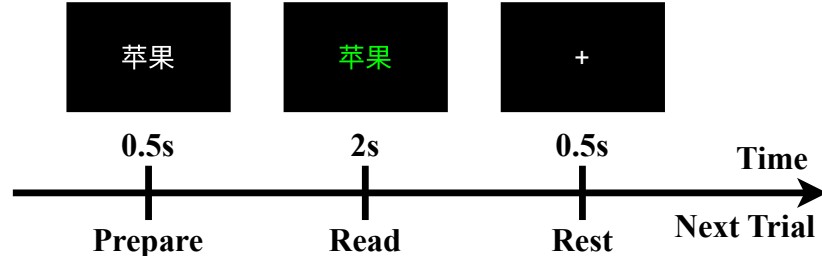

Figure 6: The experiment design of our sEEG word-reading task.

Due to the lack of open-source sEEG datasets related to language tasks, we follow the experimental design outlined by Moses et al. [37] to collect a well-annotated Chinese word-reading sEEG dataset, including 12 subjects (9 male, 3 female; aged 15-53, $\mu$ 27.8, $\sigma$ 10.4) with pharmacologically intractable epilepsy.

In the word-reading task, the subject speaks aloud individual words from a 61-word set while we simultaneously record his brain activity (measured by sEEG) and voice. The word set is chosen based on the following criteria:

- The versatility of the words in generating a range of sentences.
- The simplicity of using the words to express fundamental caregiving requirements.
- The diversity of word pronunciations to cover as many Chinese pronunciation combinations as possible.

A list of the words contained in this 61-word set is provided in Table 4.

All data are collected as a series of "blocks" (25 blocks in total), with each block lasting about 10 minutes and consisting of multiple trials. During each block of this task, all words (from the 61-word set) are presented individually twice, leading to a total of 122 trials.

Each trial in a block of this task starts with one word shown on the screen in white text. After 0.5 seconds, the text will turn green and remain on the screen for 2 seconds. This color transition from white to green represents the go cue for each trial, and the subject is instructed to speak the word aloud as soon as the text turns green. Afterward, the text will be replaced with a blank screen with a centered cross. After 0.5 seconds, the task continues to the next trial. The word presentation order is randomized within each task block.

Besides, we also collected non-task recordings of subjects in their daily life. Apart from sleep periods, there are roughly 12 hours of non-task recordings during wakefulness. In summary, for each subject, we collect about 15 hours of sEEG recordings, of which 3 hours are task recordings.

Table 4: The Chinese words and their corresponding English translations in the 61-word set.

| Words | Translations | Words | Translations | Words | Translations |
|---|---|---|---|---|---|
| 嘴巴 | mouth | 菠萝 | pineapple | 帮助 | help |
| 把 | get | 朋友 | friend | 脸盆 | washbasin |
| 平静 | calm | 漂亮 | pretty | 衣服 | clothes |
| 豆腐 | tofu | 米饭 | rice | 放在 | put on |
| 面条 | noodle | 毛巾 | towel | 关门 | close the door |
| 电脑 | computer | 凳子 | stool | 小刀 | knife |
| 头疼 | headache | 软糖 | gummies | 醋 | vinegar |
| 青菜 | vegetables | 厕所 | toilet | 葱花 | chopped green onion |
| 手机 | cell phone | 篮球 | basketball | 钢琴 | piano |
| 心情 | mood | 丝瓜 | loofah | 蒜泥 | garlic paste |
| 怎样 | how | 香肠 | sausage | 需要 | need |
| 你 | you | 拿 | hold | 橙汁 | orange juice |
| 找 | look for | 猪肉 | pork | 吃 | eat |
| 穿 | wear | 是 | be | 家人 | family |
| 热水 | hot water | 护士 | nurse | 换药 | change dressing |
| 喝 | drink | 口渴 | thirsty | 看 | look |
| 碗 | bowl | 鱼块 | steak | 感觉 | feel |
| 给 | give | 玩 | play | 问题 | problem |
| 外卖 | takeouts | 有 | have | 音乐 | music |
| 预约 | reserve | 汤圆 | sweet dumpling | 愿意 | willing |
| 我 | I | - | - | - | - |

# B  Details of Baselines

In experiments, we compare our model to the existing supervised or self-supervised methods on brain signals. The details of these baseline models are given here:

- **TS-TCC**[20]: A self-supervised model that consists only of a CNN module to capture local features. This model learns robust temporal and discriminative representations from time series by designing a tough cross-view prediction task and a contextual contrasting module. Since sEEG is a unique type of time series, this model is suitable to serve as a baseline for comparison.

- **CNN-BiGRU**[37]: A supervised model that consists of both CNN module and Bi-GRU module, to capture contextual features from EEG signals. This model is mainly designed for ECoG-based vocal production tasks, similar to ours. Since ECoG and sEEG are both intracranial neural signals of the brain, this model is suitable to serve as a baseline for comparison.

- **EEG-Conformer**[44]: A supervised model that consists of both CNN module and Transformer module, to encapsulate local and global features in a unified EEG classification framework. EEG-Conformer is mainly designed for EEG-based motor imagination tasks. Since the data modes of EEG and sEEG are similar, and the signals primarily pertain to vocal production, this model is suitable to serve as a baseline for comparison.

- **Neuro-BERT**[50]: A self-supervised model that consists of both CNN module and Transformer module, to encapsulate local and global features. This model learns robust contextual representations from EEG by introducing mask modeling. Since the data modes of EEG and sEEG are similar, this model is suitable to serve as a baseline for comparison.

- **DeWave**[19]: A supervised model that consists of both Conformer module [24] and Transformer module, to encapsulate local and global features for language decoding. We adopt its encoder, which consists of a 6-layer Conformer and a 6-layer Transformer. Then, we add a classification head, which is also used in our model, for downstream word classification. Since DeWave is also designed for language decoding, this model is suitable to serve as a baseline for comparison.

- **BrainBERT**[49]: A self-supervised model for sEEG recordings that bridges modern representation learning approaches to neuroscience. BrainBERT builds universal representation based on the superlet spectrograms of one single sEEG channel without modeling the spatial relationships among channels. Since the downstream tasks for BrainBERT are also related to language decoding (e.g., sentence-onset detection, speech vs. non-speech detection, etc.), this model is suitable to serve as a baseline for comparison.

- **Brant**[54]: A self-supervised model for sEEG recordings that can capture both long-term temporal dependency and spatial correlation from neural signals. Brant is mainly designed for medicine, serving as a sEEG foundation model. Although Brant mainly evaluates its performance on the low-level modeling tasks [51] (e.g., neural signal forecasting, imputation, etc.), Brant achieves SOTA performance on some high-level modeling tasks (e.g., seizure detection). As a foundation model in sEEG pre-training field, this model is suitable to serve as a baseline for comparison.

- **LaBraM**[28]: A self-supervised model for EEG recordings that learns generic representations with tremendous EEG data. LaBraM serves as an EEG foundation model, achieving SOTA performance on various downstream EEG tasks. Since the spatial embeddings are pre-defined according to the EEG caps, LaBraM can only be trained within one subject under the sEEG setting. Since the data modes of EEG and sEEG are similar, this model is suitable to serve as a baseline for comparison.

- **LaBraM+PopT**[28, 10]: A self-supervised model based on LaBraM, simply replacing the learnable spatial embeddings with hard-coded spatial embeddings from PopT [10] to enable multi-subject pre-training under the sEEG setting.

The detailed implementations of these baseline models are given here:

- For the TS-TCC method [20], the hyper-parameters are optimized for better performance, as they also have different hyper-parameter settings for different datasets in their original

implementation. The data samples are resampled to 400Hz. The sizes of convolution kernels are changed to {25, 8, 8} (other attempts include {8, 8, 8}, {15, 8, 8}, {20, 8, 8}, and {30, 8, 8}); the sizes of pooling kernels are changed to {10, 2, 2} (other attempts include {2, 2, 2}, {5, 2, 2}, and {20, 2, 2}); the numbers of pooling strides are changed to {10, 2, 2} (other attempts include {2, 2, 2}, {5, 2, 2}, and {20, 2, 2}). All other hyper-parameters are the same as the original implementation.

- For the CNN-BiGRU method [37], the hyper-parameters are the same as the original implementation. The data samples are resampled to the specified sampling rate.

- For the EEG-Conformer method [44], the hyper-parameters are the same as the original implementation. The data samples are resampled to the specified sampling rate.

- For the Neuro-BERT method [50], the hyper-parameters are optimized for better performance, as they also have different hyper-parameter settings for different datasets in their original implementation. The data samples are sampled to 400Hz. The sizes of convolution kernels are changed to {40,} (other attempts include {20,} and {80,}); the numbers of convolution strides are changed to {40,} (other attempts include {20,} and {80,}).

- For the DeWave method [19], the hyper-parameters are the same as the original implementation. The data samples are resampled to the specified sampling rate.

- For the BrainBERT method [49], the hyper-parameters are optimized for better performance. We change the "nperseg" and "noverlap" arguments of "scipy.signal.stft" function from {400, 350} to {1600, 1400} (other attempts include {200, 175}, {800, 700} and {3200, 2800}).

- For the Brant method [54], the hyper-parameters are optimized based on the Brant-Tiny model for better performance. We change the length of the patch segment from 6 seconds to 1 second. Besides, we change the linear embedding layer to the convolution embedding layer, which is also used in LaBraM [28]. The numbers of convolution filters are {96, 96, 96} (other attempts include {192, 192, 192}); the sizes of convolution kernels are {9, 9, 3} (other attempts include {19, 9, 3} and {9, 9, 3}); the numbers of convolution strides are {5, 5, 1} (other attempts include {10, 5, 1}) and {5, 5, 2}).

- For the LaBraM method [28], the hyper-parameters are the same as the original implementation of the LaBraM-Base model. The data samples are resampled to the specified sampling rate.

- For the LaBraM-PopT method [28, 10], the hyper-parameters are the same as the original implementation of the LaBraM-Base model. The data samples are resampled to the specified sampling rate.

When evaluating the decoding performance of these baseline models, we follow the same experiment setup of the Du-IN CLS model:

- For one subject, we split the downstream dataset into training, validation, and testing splits with a size roughly proportional to 80%, 10%, and 10%.

- The data samples are 3 seconds with the specified sampling rate corresponding to each model.

- The samples in the train-set are augmented following the pipeline defined in Appendix D.

For the self-supervised methods, the pre-training setup follows the original setup of each model:

- For the TS-TCC model, we use all sEGG recordings for each subject to pre-train it. The data samples are 4 seconds.

- For the Neuro-BERT model, we use all sEGG recordings for each subject to pre-train it. The data samples are 4 seconds.

- For the BrainBERT model, we use around 180 hours of sEEG recordings from either each subject or 12 subjects for pre-training. This pre-training dataset is larger than the one (approximately 45 hours) used in the original paper. The data samples are 4 seconds.

- For the Brant model, we also use all sEEG recordings from either each subject or 12 subjects to pre-train it. While the total pre-training dataset is smaller than the one (around 2700 hours) used in the original paper, the number of subjects (i.e., the number of sEEG location configurations) is greater than in the original paper. The data samples are 4 seconds.

- For the LaBraM model, we use all sEGG recordings for each subject to pre-train it. The data samples are 4 seconds.
- For the LaBraM-PopT model, we use all sEEG recordings from 12 subjects to pre-train it. The data samples are 4 seconds.

## C  Model Details

### C.1  Du-IN VQ-VAE

The architecture of the Du-IN VQ-VAE model contains three parts: (1) Du-IN Encoder, (2) Vector Quantizer, and (3) Du-IN Regressor. The overall architecture of "Du-IN Encoder" is shown in Figure 2. The "Vector Quantizer" is implemented similarly in LaBraM[28]. The "Du-IN Regressor" contains:

- **Transformer Decoder:** A stack of Transformer layers.
- **Time Regression Head:** A stack of 1D Transposed Convolution layers and one linear projection layer.

The hyperparameters for Du-IN VQ-VAE training are shown in Table 5.

Table 5: The hyperparameters for Du-IN VQ-VAE training.

| Module | Sub-Module | Name | Value |
|---|---|---|---|
| Du-IN Encoder | Spatial Encoder | Linear Projection | $10 \rightarrow 16$ |
| | | # of Input Channels | {16,128,128} |
| | | # of Output Channels | {128,128,16} |
| | | Kernel Size | {19,3,3} |
| | | Stride | {10,1,1} |
| | | Padding | {9,1,1} |
| | Transformer Encoder | # of Transformer Layers | 8 |
| | | Hidden Size | 160 |
| | | MLP Size | 320 |
| | | MLP Dropout Ratio | {0.2,0.} |
| | | # of Attention Heads | 8 |
| | | Attention Head Size | 64 |
| | | Attention Dropout Ratio | 0.2 |
| Vector Quantizer | - | Codex Size | $2048 \times 64$ |
| | | Embedding-to-Codex Projection | $160 \rightarrow 160(\mathrm{Tanh}) \rightarrow 64$ |
| | | Codex-to-Embedding Projection | $64 \rightarrow 160$ |
| Du-IN Regressor | Transformer Decoder | # of Transformer Layers | 4 |
| | | Hidden Size | 160 |
| | | MLP Size | 320 |
| | | MLP Dropout Ratio | {0.2,0.} |
| | | # of Attention Heads | 8 |
| | | Attention Head Size | 64 |
| | | Attention Dropout Ratio | 0.2 |
| | Time Regression Head | # of Input Channels | {160,128,128,128,128} |
| | | # of Output Channels | {128,128,128,128,16} |
| | | Kernel Size | {3,3,10,9,19} |
| | | Stride | {1,1,10,1,10} |
| | | Padding | - |
| | | Output Padding | - |
| | | Linear Projection | $16 \rightarrow 10$ |
| Optimizer | - | Batch Size | 64 |
| | | Maximum Learning Rate | 3e-4 |
| | | Minimum Learning Rate | 5e-5 |
| | | Learning Rate Scheduler | Cosine |
| | | Optimizer Type | AdamW |
| | | Adam $\beta$ | $(0.9, 0.99)$ |
| | | Weight Decay | 0.01 |
| | | Total Epochs | 400 |
| | | Warm-up Epochs | 40 |

## C.2 Du-IN MAE

The architecture of the Du-IN MAE model contains two parts: (1) Du-IN Encoder, and (2) Token Prediction Head. The overall architecture of the "Du-IN Encoder" is shown in Figure 2. The hyperparameters of "Du-IN Encoder" are the same as those in Du-IN VQ-VAE. It's worth noting that when training Du-IN MAE, the weights of the "Du-IN Encoder" are randomly initialized, instead of loaded from the pre-trained Du-IN VQ-VAE model. The hyperparameters for Du-IN MAE training are shown in Table 6.

Table 6: The hyperparameters for Du-IN MAE training.

| Module | Sub-Module | Name | Value |
|---|---|---|---|
| Token Prediction Head | - | Linear Projection | $160 \rightarrow 2048$ |
| Optimizer | - | Batch Size | 64 |
| | | Maximum Learning Rate | 3e-4 |
| | | Minimum Learning Rate | 5e-5 |
| | | Learning Rate Scheduler | Cosine |
| | | Optimizer Type | AdamW |
| | | Adam $\beta$ | $(0.9, 0.99)$ |
| | | Weight Decay | 0.05 |
| | | Total Epochs | 400 |
| | | Warm-up Epochs | 40 |

## C.3 Du-IN CLS

The architecture of the Du-IN CLS model contains two parts: (1) Du-IN Encoder, and (2) Label Prediction Head. The overall architecture of the "Du-IN Encoder" is shown in Figure 2. The hyperparameters of "Du-IN Encoder" are the same as those in Du-IN VQ-VAE. It's worth noting that the "Du-IN Encoder" weights in Du-IN CLS can be loaded from either the pre-trained Du-IN MAE or the pre-trained Du-IN VQ-VAE. In the ablation experiments shown in Table 2, our models have different suffixes:

- **Du-IN:** The original Du-IN CLS model. All weights of this model are randomly initialized.
- **Du-IN (vqvae+vq):** The weights of the "Du-IN Encoder" in the Du-IN CLS model are loaded from the pre-trained Du-IN VQ-VAE. When fine-tuning it on the downstream task, the "Vector Quantizer" in the pre-trained Du-IN VQ-VAE is inserted between "Du-IN Encoder" and "Label Prediction Head". This is the same operation in DeWave[19].
- **Du-IN (vqvae):** The weights of the "Du-IN Encoder" in the Du-IN CLS model are loaded from the pre-trained Du-IN VQ-VAE. This is the same operation in EEGFormer [12].
- **Du-IN (mae):** The weights of the "Du-IN Encoder" in the Du-IN CLS model are loaded from the pre-trained Du-IN MAE. This is the same operation in LaBraM [28].
- **Du-IN (poms):** The weights of the "Du-IN Encoder" in the Du-IN CLS model are loaded from the Du-IN MAE, which is pre-trained on multiple subjects. The modification of the Du-IN VQ-VAE and the Du-IN MAE to support multi-subject pre-training includes (1) initializing different spatial encoders for different subjects and (2) sharing the same transformer encoder and neural codex.

The "Label Prediction Head" is an MLP with one hidden layer, flattens the output embedding sequence from upstream, and maps this feature embedding to the final prediction through MLP. The hyperparameters for Du-IN CLS training are shown in Table 7.

Table 7: The hyperparameters for Du-IN CLS training.

| Module | Sub-Module | Name | Value |
|---|---|---|---|
| Label Prediction Head | - | Flatten | - |
| | | Linear Projection | $30 \times 160 \rightarrow 128(\mathrm{ReLU}) \rightarrow 61$ |
| Optimizer | - | Batch Size | 32 |
| | | Maximum Learning Rate | 2e-4 |
| | | Minimum Learning Rate | 5e-6 |
| | | Learning Rate Scheduler | Cosine |
| | | Optimizer Type | AdamW |
| | | Adam $\beta$ | $(0.9, 0.99)$ |
| | | Weight Decay | 0.05 |
| | | Total Epochs | 200 |
| | | Warm-up Epochs | 20 |

# D    Data Augmentation

To enhance the robustness of learned representations during both the pre-training and fine-tuning stages, we apply data augmentation in both datasets.

**Pre-training Dataset.**    In our implementation, we segment sEEG recordings into 8-second samples with a 4-second overlap. When fetching a sample, we randomly select a starting point between 0 and 4 seconds, then extract a 4-second sample beginning from that point.

**Downstream Dataset.**    Since a trial lasts for 3 seconds, employing the jittering mentioned above leads to the blending of information from other trials. In our implementation, we segment sEEG recordings into 3-second samples. When fetching a sample, we randomly choose a shift step between 0 and 0.3 seconds, then shift the sample either to the left or right, padding it with zeros.

# E    Du-IN Pre-training Analysis

The pre-training of Du-IN can be interpreted as the training of a variational autoencoder [30, 3]. Let $x$ denote the original sEEG signal, $\tilde{x}$ the corrupted sEEG by mask, and $z$ the neural tokens. Considering the evidence lower bound (ELBO) of the log-likelihood $p(x|\tilde{x})$, i.e., recovering the original sEEG signal from its corrupted version:

$$\sum_{(x_i,\tilde{x}_i)\in\mathcal{D}} \log p(x_i|\tilde{x}_i) \geq \sum_{(x_i,\tilde{x}_i)\in\mathcal{D}} \Big( \underbrace{\mathbb{E}_{z_i\sim q_\phi(\mathbf{z}|x_i)}[\log p_\psi(x_i|z_i)]}_{\text{Neural Token Reconstruction}} - D_{\text{KL}}[q_\phi(\mathbf{z}|x_i), p_\theta(\mathbf{z}|\tilde{x}_i)] \Big), \quad (10)$$

where (1) $q_\phi(z|x)$ denotes the Du-IN Encoder in the Du-IN VQ-VAE that obtains neural tokens; (2) $p_\psi(x|z)$ decodes the original sEEG signal given input neural tokens; (3) $p_\theta(z|\tilde{x})$ recovers the neural tokens based on the masked sEEG signal, which is our Du-IN pre-training task.

The whole framework is optimized through a two-stage procedure as [47, 40]. For the first stage, we train the Du-IN Encoder in the Du-IN VQ-VAE as a discrete variational autoencoder by minimizing the reconstruction loss $-\mathbb{E}_{z_i\sim q_\phi(\mathbf{z}|x_i)}\log p_\psi(\tilde{x}_i|z_i)$ with a uniform prior. For the second stage, we set $q_\phi$ as well as $p_\psi$ fixed and learn the prior $p_\theta$ by minimizing the loss $D_{\text{KL}}$. For simplicity, $q_\phi(\mathbf{z}|x_i)$ is defined as a one-point distribution with the most likely neural tokens $\hat{z}_i = \arg\max_z q_\phi(\mathbf{z}|x_i)$.

Consequently, we can rewrite Equation 10 as

$$\sum_{(x_i,\tilde{x}_i)\in\mathcal{D}} \log p(x_i|\tilde{x}_i) \geq \sum_{(x_i,\tilde{x}_i)\in\mathcal{D}} \Big( \underbrace{\mathbb{E}_{z_i\sim q_\phi(\mathbf{z}|x_i)}[\log p_\psi(\tilde{x}_i|z_i)]}_{\text{Neural Token Reconstruction}} + \underbrace{\log p_\theta(\hat{z}_i|\tilde{x}_i)}_{\text{Masked sEEG Modeling}} \Big), \quad (11)$$

where the first term is the objective for vector-quantized neural signal regression in the first stage (i.e., the Du-IN VQ-VAE model), and the second term is the objective for Du-IN pre-training in the second stage (i.e., the Du-IN MAE model).

# F    Visualization of Vector-Quantized sEEG Regression

We further visualize how the sEEG signals are reconstructed. As depicted in Figure 7, although some details are missing, the overall trend of the signals is reconstructed well. Meanwhile, there is a stable decrease in the reconstruction loss during training, which indicates the discrete codex does learn high-level information from sEEG signals.

# G    Visualization of Mask sEEG Modeling

Figure 8 demonstrates the convergence curves of the total pre-training loss and masked sEEG modeling accuracy of the Du-IN MAE model. We observe that there is a stable decrease in the mask modeling loss, and the mask modeling accuracy achieves about 20%, which is similar to [28].

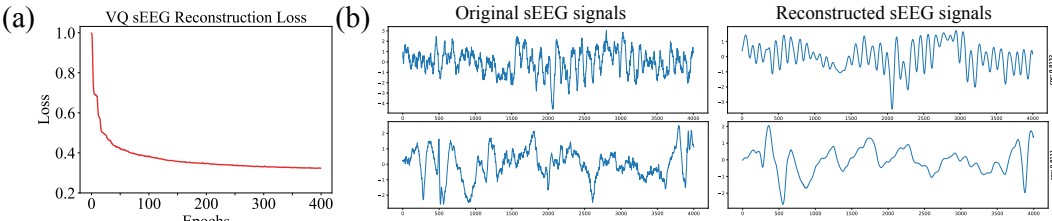

Figure 7: **The visualization of Vector-Quantized sEEG Regression. (a).** The reconstruction loss curve during the training process of the Du-IN VQ-VAE model. **(b).** The visualization of reconstructed sEEG signals.

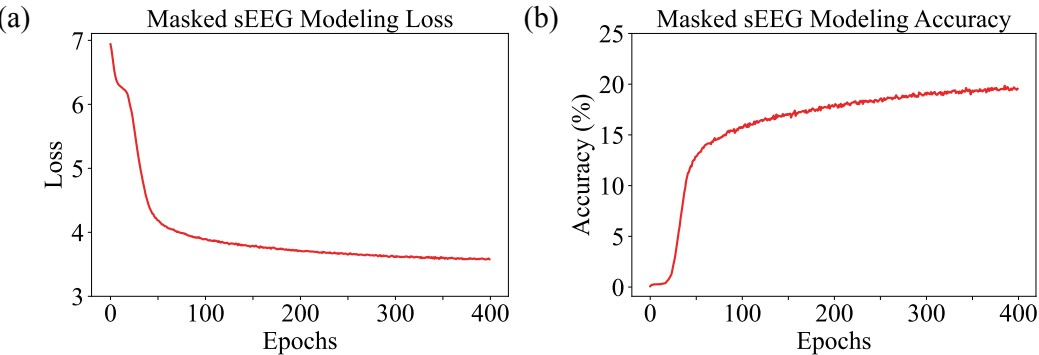

Figure 8: The loss curve and accuracy curve during the training process of the Du-IN MAE model.

## H  Channel Contribution Analysis

For each subject, after training the Du-IN model (with random initialization) on the downstream dataset, we utilize the weights $W \in \mathbb{R}^{C \times D}$ of linear projection in the spatial encoder to calculate the contribution scores $\mathcal{S}$ of channels:

$$\mathcal{S} = \{s_i | i = 1, ..., C\}, \quad s_i = \frac{1}{D} \sum_{j=1}^{D} |W_{ij}|, \tag{12}$$

where $C$ is the number of channels, $D$ is the dimension of projected embedding and $|\cdot|$ gets the absolute value. Then, we normalize $\mathcal{S}$ using its maximum value to ensure it falls within the [0,1] range. Finally, given the variability in model performance across subjects, we further adjust the channel contribution scores based on the decoding performance of that subject, i.e., $\mathcal{S} = \{s_i \cdot p | i = 1, ..., C\}$, where $p$ represents the decoding performance of that subject.

After calculating the channel contribution scores of all subjects, we project them to the standard brain template according to the MNI (Montreal Neurological Institute) locations of channels, using Nilearn 0.9.2. Since the electrodes are sparsely distributed within the brain, we use Scipy 1.8.1 to interpolate and smooth the channel contribution matrix and use NiLearn to plot the channel contribution map demonstrated in Figure 4 (a).

With the sorted channels within each subject, we evaluate the effect of the number of channels on the decoding performance. For each subject, we evaluate the Du-IN model with $\{5, 10, 15, 20, 30, 60\}$ channels (sorted by channel contribution scores), and the averaged performance (across subjects) is demonstrated in Figure 4 (b).

# I  Effectiveness of Region-Specific Channel Selection

DeWave [19] successfully reconstructs 128-channel EEG signals with the same setting of vector-quantizer. However, this is not the case under the sEEG setting, which is shown in Table 8. It's worth noting that sEEG signals are fundamentally different from EEG signals due to (1) the high information density and (2) the high specificity of different regions. Due to the desynchronization nature [7] of the brain during awake tasks, only specific brain regions are related to tasks. Therefore, only after region-specific channel selection, the Du-IN VQ-VAE model can successfully reconstruct the original signals, thus identifying the fine-grained state of brain regions.

Table 8: Ablations to validate the effectiveness of region-specific channel selection.

| Settings | Setting 1 | Setting 2 | Setting 3 |
|---|---|---|---|
| **MSE** | **0.2969±0.0376** | 0.5211±0.0492 | 0.9673±0.0148 |

[1] Setting 1: Select top-10 channels relevant to speech decoding for neural signal reconstruction.
[2] Setting 2: Randomly select 10 channels for neural signal reconstruction.
[3] Setting 3: Use all channels (109.75 channels on average) for neural signal reconstruction.

# J  Additional Group-Wise Evaluation

The cross-entropy loss of different methods from each subject is provided in Table 9, with the best in **bold** and the second underlined. For model comparison, we report the average and standard deviation values (within each subject) on six different random seeds to obtain comparable results. "Std" means standard deviation.

Table 9: The cross-entropy loss of different methods (with the best in **bold** and the second underlined).

| Methods | Token Level | Config PT[1] | Config MS[2] | Model Size | Cross-Entropy ± Ste |
|---|---|---|---|---|---|
| TS-TCC[20] | Region | ✓ | ✗ | 0.32M | 3.8871±0.3072 |
| CNN-BiGRU[37] | Region | ✗ | - | 0.54M | 4.0294±0.7621 |
| EEG-Conformer[44] | Region | ✗ | - | 2.34M | 3.8165±0.3456 |
| Neuro-BERT[50] | Region | ✓ | ✗ | 2.14M | 3.6416±0.4360 |
| DeWave[19] | Region | ✗ | - | 5.70M | 4.1891±0.5722 |
| BrainBERT[49] | Channel | ✓ | ✗ | 43.58M | 4.6254±0.1984 |
| BrainBERT[49] | Channel | ✓ | ✓ | 43.58M | 4.6190±0.2132 |
| Brant[54] | Channel | ✓ | ✗ | 69.35M | 4.7962±0.7082 |
| Brant[54] | Channel | ✓ | ✓ | 69.35M | 5.0294±1.0621 |
| LaBraM[28] | Channel | ✓ | ✗ | 6.85M | 4.8591±0.2723 |
| LaBraM-PopT[28, 10] | Channel | ✓ | ✓ | 6.85M | 4.6564±0.1893 |
| Du-IN | Region | ✗ | - | 4.38M | 3.5083±0.3003 |
| Du-IN (vqvae+vq) | Region | ✓ | ✗ | 4.38M | 3.7244±0.3104 |
| Du-IN (vqvae) | Region | ✓ | ✗ | 4.38M | 3.4309±0.2781 |
| Du-IN (mae) | Region | ✓ | ✗ | 4.38M | **3.3707±0.2882** |
| Du-IN (poms) | Region | ✓ | ✓ | 5.18M | 3.4429±0.2754 |

[1] PT: Whether the model is pre-trained before evaluation.
[2] MS: Whether the model is pre-trained across multiple subjects.

# K Subject-Wise Evaluation

The detailed performance of different methods from each subject is provided in Table 10, Table 11, and Table 12, with the best in **bold** and the second underlined. For model comparison, we report the average and standard deviation values (within each subject) on six different random seeds to obtain comparable results. "Std" means standard deviation.

Table 10: The performance of different methods from subjects (01-04).

| Methods | Config PT[1] | MS[2] | Accuracy (%) ± Std (%) subj-01 | subj-02 | subj-03 | subj-04 |
|---|---|---|---|---|---|---|
| TS-TCC[20] | ✓ | ✗ | 26.90±1.72 | 61.57±1.21 | 6.65±1.09 | 20.66±0.79 |
| CNN-BiGRU[37] | ✗ | - | 46.46±4.03 | 68.06±1.56 | 4.35±0.44 | 17.68±3.88 |
| EEG-Conformer[44] | ✗ | - | 58.41±1.03 | 69.82±1.22 | 19.50±1.71 | 49.65±2.38 |
| Neuro-BERT[50] | ✓ | ✗ | 60.44±2.23 | 72.97±1.47 | 28.38±4.23 | 52.76±3.41 |
| DeWave[19] | ✗ | - | 43.31±5.80 | 57.12±3.01 | 3.70±5.35 | 33.52±3.98 |
| BrainBERT[49] | ✓ | ✗ | 5.30±0.90 | 23.49±1.29 | 2.74±0.40 | 5.18±0.38 |
| BrainBERT[49] | ✓ | ✓ | 6.76±0.64 | 25.64±1.23 | 2.97±0.66 | 5.09±0.44 |
| Brant[54] | ✓ | ✗ | 8.64±1.59 | 47.97±1.30 | 3.06±0.36 | 2.74±0.68 |
| Brant[54] | ✓ | ✓ | 7.47±2.83 | 54.26±1.63 | 3.34±0.38 | 4.15±1.35 |
| LaBraM[28] | ✓ | ✗ | 12.55±1.17 | 39.20±1.53 | 3.54±0.47 | 13.30±1.19 |
| LaBraM-PopT[28, 10] | ✓ | ✓ | 14.14±1.28 | 39.50±1.35 | 3.28±0.55 | 12.77±1.72 |
| Du-IN | ✗ | - | 71.25±1.44 | 77.99±0.87 | 23.04±4.76 | 59.91±4.58 |
| Du-IN (vqvae+vq) | ✓ | ✗ | 50.15±3.80 | 62.79±4.67 | 20.72±2.15 | 48.24±2.65 |
| Du-IN (vqvae) | ✓ | ✗ | 72.36±1.55 | 79.16±1.12 | 29.21±2.38 | 63.83±1.83 |
| Du-IN (mae) | ✓ | ✗ | **78.60±0.79** | **83.61±0.38** | **38.80±2.52** | **70.98±0.81** |
| Du-IN (poms) | ✓ | ✓ | 73.23±0.67 | 79.12±1.07 | 28.89±1.96 | 64.24±1.29 |

[1] PT: Whether the model is pre-trained before evaluation.
[2] MS: Whether the model is pre-trained across multiple subjects.

Table 11: The performance of different methods from subjects (05-08).

| Methods | Config | | Accuracy (%) ± Std (%) | | | |
|---------|--------|---------|--------|--------|--------|--------|
| | PT[1] | MS[2] | subj-05 | subj-06 | subj-07 | subj-08 |
| TS-TCC[20] | ✓ | ✗ | 34.53±1.34 | 9.73±0.97 | 24.83±1.15 | 20.08±1.60 |
| CNN-BiGRU[37] | ✗ | - | 51.26±4.93 | 31.52±1.48 | 47.75±1.12 | 24.64±4.44 |
| EEG-Conformer[44] | ✗ | - | 65.44±1.31 | 31.06±2.58 | 47.89±1.86 | 42.12±2.08 |
| Neuro-BERT[50] | ✓ | ✗ | 71.61±1.97 | 36.63±2.15 | 50.23±2.63 | 43.24±1.36 |
| DeWave[19] | ✗ | - | 45.20±3.08 | 26.88±1.92 | 38.24±2.15 | 29.15±3.28 |
| BrainBERT[49] | ✓ | ✗ | 9.59±0.90 | 2.67±0.31 | 4.79±0.64 | 5.10±0.59 |
| BrainBERT[49] | ✓ | ✓ | 11.28±1.17 | 3.00±0.47 | 5.31±0.55 | 5.22±0.76 |
| Brant[54] | ✓ | ✗ | 24.10±2.14 | 5.09±1.27 | 7.74±1.46 | 8.66±1.07 |
| Brant[54] | ✓ | ✓ | 28.83±1.93 | 5.28±1.48 | 9.70±1.85 | 8.93±1.39 |
| LaBraM[28] | ✓ | ✗ | 15.52±1.48 | 6.63±0.33 | 12.65±0.50 | 7.41±0.86 |
| LaBraM-PopT[28, 10] | ✓ | ✓ | 17.73±1.11 | 5.81±1.61 | 12.90±1.26 | 6.41±1.85 |
| Du-IN | ✗ | - | 77.60±1.20 | 41.91±1.80 | 59.63±2.20 | 52.35±2.18 |
| Du-IN (vqvae+vq) | ✓ | ✗ | 63.46±2.28 | 34.84±1.98 | 45.20±2.44 | 40.14±1.77 |
| Du-IN (vqvae) | ✓ | ✗ | 78.56±1.24 | 43.29±1.67 | 62.29±1.49 | 54.10±1.34 |
| Du-IN (mae) | ✓ | ✗ | **81.56±1.11** | 46.90±1.02 | **65.45±1.74** | **59.09±0.98** |
| Du-IN (poms) | ✓ | ✓ | 76.91±1.38 | **47.01±1.76** | 61.68±0.95 | 55.17±1.37 |

[1] PT: Whether the model is pre-trained before evaluation.
[2] MS: Whether the model is pre-trained across multiple subjects.

Table 12: The performance of different methods from subjects (09-12).

| Methods | Config | | Accuracy (%) ± Std (%) | | | |
|---------|--------|---------|--------|--------|--------|--------|
| | PT[1] | MS[2] | subj-09 | subj-10 | subj-11 | subj-12 |
| TS-TCC[20] | ✓ | ✗ | 37.75±1.22 | 5.71±0.38 | 35.72±0.95 | 14.12±0.68 |
| CNN-BiGRU[37] | ✗ | - | 44.03±5.88 | 7.11±0.71 | 28.44±3.42 | 13.17±3.41 |
| EEG-Conformer[44] | ✗ | - | 56.51±1.98 | 22.22±1.07 | 57.10±2.03 | 29.87±1.44 |
| Neuro-BERT[50] | ✓ | ✗ | 54.12±4.11 | 24.66±1.28 | 62.99±0.93 | 36.07±2.36 |
| DeWave[19] | ✗ | - | 41.98±4.60 | 6.22±0.94 | 44.60±3.35 | 19.22±2.95 |
| BrainBERT[49] | ✓ | ✗ | 6.82±1.42 | 2.55±0.49 | 8.73±0.92 | 3.71±1.02 |
| BrainBERT[49] | ✓ | ✓ | 7.20±1.37 | 2.49±0.43 | 10.60±1.22 | 4.41±0.88 |
| Brant[54] | ✓ | ✗ | 6.82±1.44 | 2.84±0.26 | 8.76±1.55 | 7.53±1.29 |
| Brant[54] | ✓ | ✓ | 6.46±1.66 | 3.00±0.31 | 9.82±1.71 | 7.82±1.66 |
| LaBraM[28] | ✓ | ✗ | 8.97±0.52 | 3.50±0.30 | 7.92±0.61 | 7.19±0.54 |
| LaBraM-PopT[28, 10] | ✓ | ✓ | 9.35±1.09 | 3.91±0.31 | 7.84±0.92 | 7.74±1.24 |
| Du-IN | ✗ | - | 66.39±0.47 | 27.07±2.24 | 73.56±1.09 | 44.76±3.74 |
| Du-IN (vqvae+vq) | ✓ | ✗ | 60.06±1.61 | 22.05±1.76 | 50.31±4.69 | 32.06±3.28 |
| Du-IN (vqvae) | ✓ | ✗ | 67.18±1.22 | 31.06±1.59 | 72.41±1.98 | 45.38±2.26 |
| Du-IN (mae) | ✓ | ✗ | 69.18±1.96 | 34.23±1.17 | **75.52±1.27** | **48.54±0.56** |
| Du-IN (poms) | ✓ | ✓ | **70.71±1.48** | **36.90±1.34** | 72.80±1.38 | 43.53±3.20 |

[1] PT: Whether the model is pre-trained before evaluation.
[2] MS: Whether the model is pre-trained across multiple subjects.

## L    Effectiveness of Vector-Quantized Neural Signal Prediction

To verify the effectiveness of vector-quantized neural signal prediction, we elaborate on two types of experimental settings as illustrated in Table 13. The comparison between Du-IN and Setting 1 demonstrates that the codex is effective for masked sEEG modeling. The comparison between Du-IN and Setting 2 demonstrates that introducing the codex can prevent the model from focusing too much on reconstructing details, thus enabling the Du-IN MAE to learn better contextual embeddings.

Table 13: Ablations to validate the effectiveness of vector-quantized neural signal prediction.

| Model | Du-IN (mae) | Setting 1 | Setting 2 |
|---|---|---|---|
| Acc. (%) $\pm$ Ste (%) | **62.70$\pm$4.69** | 60.92$\pm$4.38 | 58.72$\pm$5.02 |

[1] Setting 1: We directly predict output embeddings of the Du-IN Encoder in the Du-IN VQ-VAE by maximizing cosine similarity instead of predicting the discrete neural tokens from the codex.
[2] Setting 2: We discard the Du-IN Encoder in the Du-IN VQ-VAE and directly reconstruct raw EEG patches by minimizing MSE loss.

## M    Ablation on Mask Ratio

In this experiment, we conduct different mask ratio settings to explore its impact. It is noted that we introduce the symmetric masking strategy, so we only need to validate half of the mask ratios. As the mask ratio is set to $r$, the symmetric masking will mask $1 - r$ proportion of sEEG patches. The ablation results are provided in Table 14. It can be induced that the best mask ratio is 0.5 (0.5) for our dataset.

Table 14: Ablations to explore the impact of mask ratios.

| Mask Ratio | 0.5 (0.5) | 0.4 (0.6) | 0.3 (0.7) | 0.2 (0.8) | 0.1 (0.9) |
|---|---|---|---|---|---|
| Acc. (%) $\pm$ Ste (%) | **62.70$\pm$4.69** | 60.58$\pm$4.33 | 59.58$\pm$4.98 | 58.92$\pm$4.07 | 58.55$\pm$3.94 |

## N    Ablation on Pre-training Epochs

The impact of the number of pre-training epochs (of the Du-IN VQ-VAE model) is demonstrated in Table 15. We use the checkpoints according to the specified epochs to pre-train the Du-IN MAE model for 400 epochs. Once the reconstruction loss of the Du-IN VQ-VAE model converges, the Du-IN VQ-VAE model can extract the state of the brain region well, thus leading to better performance.

The impact of the number of pre-training epochs (of the Du-IN MAE model) is demonstrated in Table 16. We use the checkpoints according to the specified epochs for downstream classification. Once the mask modeling loss of the Du-IN MAE model converges, the Du-IN MAE model learns robust contextual embeddings, thus leading to better performance.

Table 15: Ablations to explore the impact of the pre-training epochs (of the Du-IN VQ-VAE model).

| # of Epochs | 5 | 10 | 50 | 100 | 400 |
|---|---|---|---|---|---|
| Acc. (%) $\pm$ Ste (%) | 50.02$\pm$4.91 | 52.29$\pm$5.09 | 61.09$\pm$4.28 | 62.59$\pm$4.32 | **62.70$\pm$4.69** |

Table 16: Ablations to explore the impact of the pre-training epochs (of the Du-IN MAE model).

| # of Epochs | 5 | 10 | 50 | 100 | 400 |
|---|---|---|---|---|---|
| Acc. (%) $\pm$ Ste (%) | 57.87$\pm$4.58 | 58.12$\pm$4.49 | 61.89$\pm$4.62 | 62.47$\pm$4.77 | **62.70$\pm$4.69** |

## O   Subject-Wise Electrode Locations

We provide detailed information on the locations of the implanted sEEG electrodes for each subject. Red channels are the top 10 channels (selected through channel contribution analysis) for both pre-training and downstream evaluation, as described in Section 4.3. As the majority of subjects have sEEG electrodes implanted on only one side of their brains to locate the source of epilepsy, we provide side views of either the left or right brain areas here.

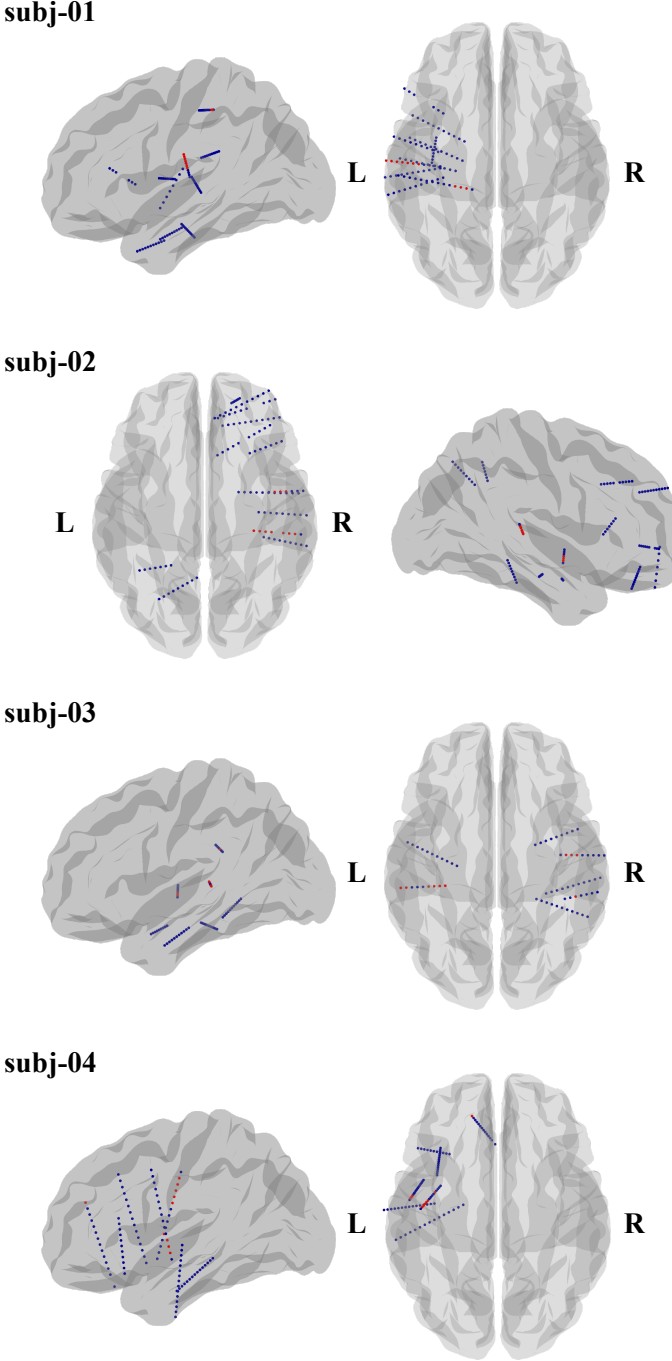

Figure 9: Electrode locations from subjects (01-04).

**subj-05**

**subj-06**

**subj-07**

**subj-08**

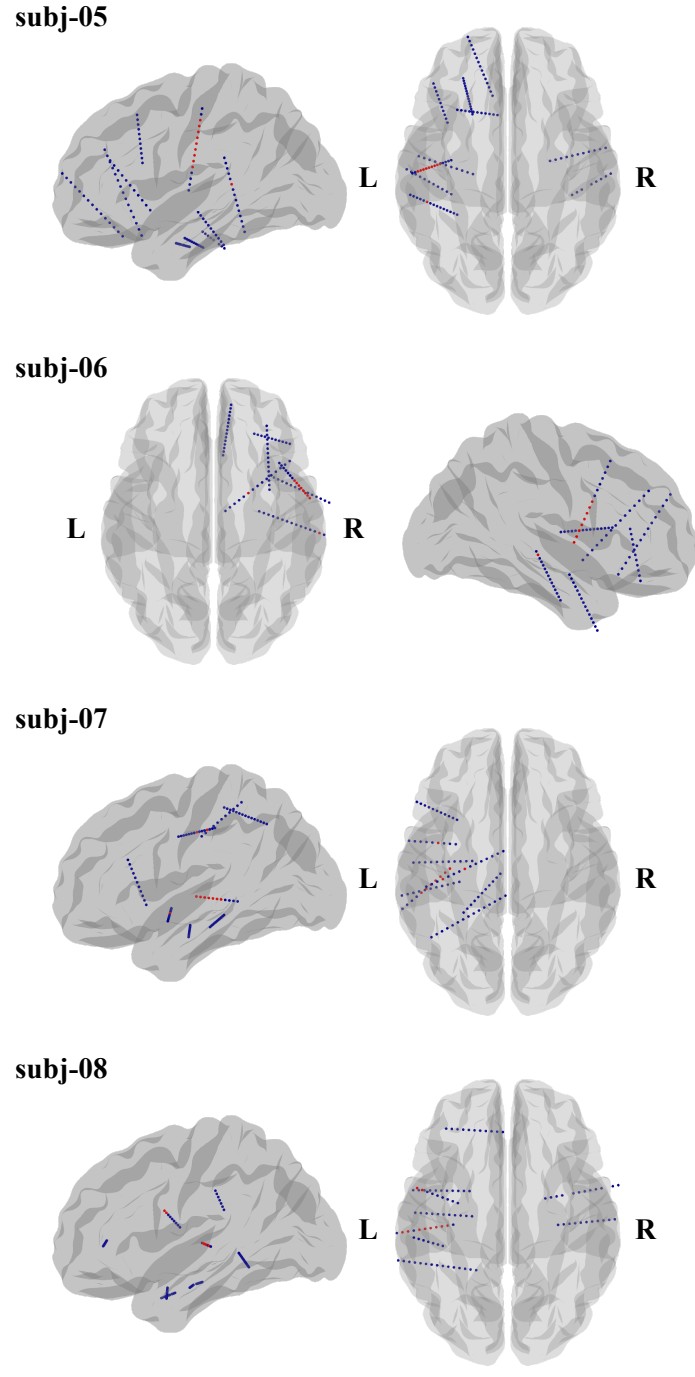

Figure 10: Electrode locations from subjects (05-08).

**subj-09**

**subj-10**

**subj-11**

**subj-12**

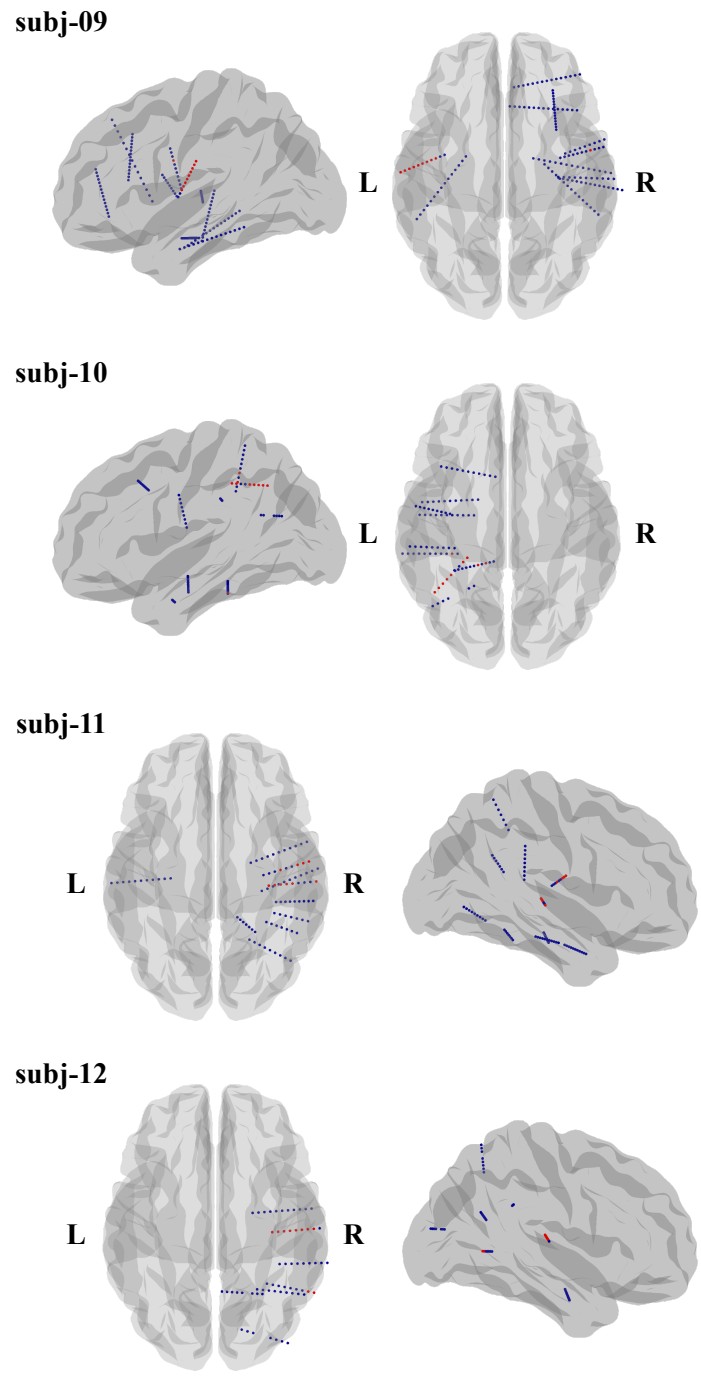

Figure 11: Electrode locations from subjects (09-12).

# P   Subject-Wise Selected Channels

The MNI coordinates, and brain region labels (according to Harvard-Oxford cortical and subcortical structural atlases [15]) for selected channels are listed below. The channels for each subject are arranged in descending order based on their contribution scores.

Table 17: The MNI coordinates and brain region labels of selected channels from subjects (01-04).

| Subjects | MNI coordinate | | | Brain Region | Du-IN Accuracy (%) |
|---|---|---|---|---|---|
| | x | y | z | | |
| subj-01 | -57 | -16 | 19 | Central Opercular Cortex L | 71.25 |
| | -61 | -16 | 20 | Postcentral Gyrus L | |
| | -54 | -16 | 17 | Central Opercular Cortex L | |
| | -67 | -15 | 23 | Postcentral Gyrus L | |
| | -25 | -30 | 49 | White L | |
| | -64 | -15 | 21 | Postcentral Gyrus L | |
| | -51 | -17 | 16 | Central Opercular Cortex L | |
| | -22 | -31 | 49 | White L | |
| | -48 | -17 | 15 | Central Opercular Cortex L | |
| | -18 | -32 | 49 | White L | |
| subj-02 | 33 | -27 | 7 | White R | 77.99 |
| | 37 | -28 | 7 | Heschls Gyrus R | |
| | 30 | -27 | 6 | White R | |
| | 48 | -29 | 9 | Planum temporale R | |
| | 51 | -29 | 10 | Planum temporale R | |
| | 41 | -28 | 8 | White R | |
| | 46 | -3 | -9 | Insula R | |
| | 55 | -29 | 11 | Planum temporale R | |
| | 49 | -3 | -7 | Planum temporale R | |
| | 43 | -3 | -10 | Insula R | |
| subj-03 | -38 | -30 | 6 | White L | 23.04 |
| | -58 | -31 | 4 | Superior Temporal Gyrus L | |
| | -31 | -30 | 7 | White L | |
| | -35 | -30 | 7 | White L | |
| | 49 | -11 | 1 | Heschls Gyrus R | |
| | 48 | -36 | 26 | Parietal Operculum Cortex R | |
| | 42 | -11 | -1 | Insula R | |
| | -55 | -31 | 5 | Superior Temporal Gyrus L | |
| | -41 | -30 | 6 | Planum temporale L | |
| | 45 | -11 | 0 | Heschls Gyrus R | |
| subj-04 | -44 | -10 | 32 | Precentral Gyrus L | 59.91 |
| | -45 | -11 | 35 | Precentral Gyrus L | |
| | -53 | -6 | -1 | Planum temporale L | |
| | -44 | -9 | 28 | Precentral Gyrus L | |
| | -46 | -12 | 39 | Precentral Gyrus L | |
| | -52 | -5 | 2 | Planum temporale L | |
| | -43 | -8 | 24 | White L | |
| | -38 | -3 | 6 | Insula L | |
| | -53 | -7 | -5 | Superior Temporal Gyrus L | |
| | -16 | 43 | 25 | White L | |

Table 18: The MNI coordinates and brain region labels of selected channels from subjects (05-08).

| Subjects | MNI coordinate | | | Brain Region | Du-IN Accuracy (%) |
|---|---|---|---|---|---|
| | x | y | z | | |
| subj-05 | -48 | -16 | 33 | Postcentral Gyrus L | 77.60 |
| | -46 | 15 | 29 | Postcentral Gyrus L | |
| | -43 | -14 | 22 | White L | |
| | -51 | -17 | 39 | Postcentral Gyrus L | |
| | -44 | -15 | 26 | White L | |
| | -53 | -17 | 43 | Postcentral Gyrus L | |
| | -50 | -16 | 36 | Postcentral Gyrus L | |
| | -41 | -14 | 19 | Insula L | |
| | -55 | -18 | 46 | Postcentral Gyrus L | |
| | -49 | -37 | 9 | White L | |
| subj-06 | 56 | -1 | 10 | Central Opercular Cortex R | 41.91 |
| | 58 | -4 | 4 | Planum temporale R | |
| | 52 | 4 | 21 | Precentral Gyrus R | |
| | 53 | 3 | 17 | Precentral Gyrus R | |
| | 57 | -2 | 7 | Central Opercular Cortex R | |
| | 54 | 1 | 14 | Precentral Gyrus R | |
| | 51 | 6 | 24 | Precentral Gyrus R | |
| | 64 | -25 | -4 | Middle Temporal Gyrus R | |
| | 21 | -1 | 11 | White R | |
| | 49 | 7 | 28 | Precentral Gyrus R | |
| subj-07 | -38 | -18 | 2 | Insula L | 59.63 |
| | -44 | -23 | 1 | Heschls Gyrus L | |
| | -41 | -21 | 1 | Heschls Gyrus L | |
| | -35 | -16 | 2 | Insula L | |
| | -50 | -28 | 0 | Superior Temporal Gyrus L | |
| | -47 | -26 | 1 | Planum temporale L | |
| | -52 | -30 | 0 | Superior Temporal Gyrus L | |
| | -26 | -16 | 40 | White L | |
| | -42 | 0 | -8 | Insula L | |
| | -39 | -22 | 41 | Postcentral Gyrus L | |
| subj-08 | -40 | -20 | 4 | Heschls Gyrus L | 52.35 |
| | -43 | -21 | 4 | Heschls Gyrus L | |
| | -37 | -20 | 4 | Insula L | |
| | -50 | -22 | 4 | Heschls Gyrus L | |
| | -47 | -21 | 4 | Heschls Gyrus L | |
| | -55 | 3 | 24 | Precentral Gyrus L | |
| | -64 | -24 | 3 | Superior Temporal Gyrus L | |
| | -61 | -23 | 3 | Superior Temporal Gyrus L | |
| | -54 | -22 | 3 | Planum temporale L | |
| | -52 | 2 | 22 | Planum temporale L | |

Table 19: The MNI coordinates and brain region labels of selected channels from subjects (09-12).

| Subjects | MNI coordinate | | | Brain Region | Du-IN Accuracy (%) |
|---|---|---|---|---|---|
| | x | y | z | | |
| subj-09 | -58 | -13 | 21 | Postcentral Gyrus L | 66.39 |
| | -55 | -12 | 19 | Central Opercular Cortex L | |
| | -53 | -11 | 17 | Central Opercular Cortex L | |
| | -50 | -10 | 15 | Central Opercular Cortex L | |
| | -61 | -15 | 23 | Postcentral Gyrus L | |
| | -63 | -16 | 25 | Postcentral Gyrus L | |
| | -45 | -8 | 10 | Central Opercular Cortex L | |
| | -47 | -9 | 12 | Central Opercular Cortex L | |
| | -42 | -7 | 8 | Insula L | |
| | 52 | -2 | 26 | Precentral Gyrus R | |
| subj-10 | -34 | -47 | 41 | Supramarginal Gyrus L | 27.07 |
| | -42 | -55 | 41 | Angular Gyrus L | |
| | -39 | -53 | 41 | Angular Gyrus L | |
| | -37 | -50 | 41 | Supramarginal Gyrus L | |
| | -25 | -37 | 42 | White L | |
| | -44 | -58 | 41 | Angular Gyrus L | |
| | -27 | -39 | 42 | White L | |
| | -18 | -41 | 48 | Postcentral Gyrus L | |
| | -34 | -34 | -23 | Temporal Fusiform Cortex L | |
| | -13 | -40 | 43 | Precuneous Cortex L | |
| subj-11 | 39 | -22 | 3 | Heschls Gyrus R | 73.56 |
| | 36 | -23 | 3 | Insula R | |
| | 47 | -22 | 2 | White R | |
| | 32 | -23 | 4 | White R | |
| | 43 | -22 | 2 | Planum temporale R | |
| | 53 | -9 | 16 | Central Opercular Cortex R | |
| | 57 | -8 | 17 | Postcentral Gyrus R | |
| | 50 | -10 | 16 | Central Opercular Cortex R | |
| | 61 | -21 | 0 | Superior Temporal Gyrus R | |
| | 39 | -14 | 13 | Insula R | |
| subj-12 | 45 | -20 | 11 | Heschls Gyrus R | 44.76 |
| | 38 | -20 | 12 | Insula R | |
| | 42 | -20 | 11 | Heschls Gyrus R | |
| | 49 | -20 | 10 | Heschls Gyrus R | |
| | 53 | -19 | 10 | Heschls Gyrus R | |
| | 60 | -19 | 9 | Planum temporale R | |
| | 56 | -19 | 9 | Planum temporale R | |
| | 60 | -58 | 3 | Middle Temporal Gyrus R | |
| | 56 | -57 | 3 | Middle Temporal Gyrus R | |
| | 35 | -21 | 12 | Insula R | |

