# OpenReview forum: "Du-IN: Discrete units-guided mask modeling for decoding speech from Intracranial Neural signals"
_NeurIPS.cc/2024/Conference — NeurIPS 2024 poster_

### Official Review · Reviewer_t1mV · 2024-07-10

**Soundness:** 2
**Presentation:** 3
**Contribution:** 2
**Rating:** 4
**Confidence:** 5

**Summary:**

The authors proposed Du-IN, a speech decoding framework that incorporates a self-supervised pre-training stage and a classification stage. Experiments are conducted on 12 subjects with a task of word classification. Comparisons are conducted to demonstrate effectiveness.

**Strengths:**

* (a) The paper is clearly written and easy to follow.

* (b) The task of speech decoding using intracranial recordings is interesting.

**Weaknesses:**

* (a) Although the task of decoding speech from intracranial recordings is an interesting topic, the algorithmic novelty of this paper is limited. The pretraining framework the authors proposed is very similar to BEIT other than the underlying data modality being images and sEEG. Simply applying an existing model BEIT to sEEG data with minor modifications to the backbone architecture does not meet the high standards required for acceptance at a top conference like NeurIPS.

* (b) The comparisons conducted are insufficient. In particular, besides BrainBERT and Brant, the authors compared with, there are also many pre-training approaches for neurological signals.  I list some more up-to-date approaches from a contrastive learning perspective and a masked autoencoding perspective. For contrastive-based approaches,  there are TS-TCC [1] and [2]. For masked autoencoding based approaches, there are Biot [3] and Neuro-BERT [4]. As a matter of fact, the authors could at least follow the comparison setup in Brant for a fair comparison.

* (c) Reproduction details are missing. The reported results on BrainBERT and Brant are ~7.5% and ~12.5%, as compared to the reported performance of ~63% of the proposed  Du-IN. It is astonishing that the proposed Du-IN outperforms BrainBERT and Brant by almost 50%. Both BrainBERT and Brant are masked autoregressive based pertaining approaches. Although BrainBERT and Brant vary from Du-IN, it does not make sense that pretraining could lead to such a tremendous difference. The authors should report the hyperparameters used to reproduce Brant and BrainBERT. Otherwise, It is highly possible that the authors did not tune  BrainBERT and Brant properly.

* (d) Although the decoding of speech is an interesting question, it seems rather well-established in open vocabulary scenarios [5][6], decoding words of limited vocabulary traces back to at least ten years ago.




**References**

[1] Woo, Gerald, et al. "Cost: Contrastive learning of disentangled seasonal-trend representations for time series forecasting." arXiv preprint arXiv:2202.01575 (2022).

[2] Baevski, Alexei, et al. "Data2vec: A general framework for self-supervised learning in speech, vision, and language." International Conference on Machine Learning. PMLR, 2022.

[3] Yang, Chaoqi, M. Westover, and Jimeng Sun. "Biot: Biosignal transformer for cross-data learning in the wild." Advances in Neural Information Processing Systems 36 (2024).

[4] Wu, Di, et al. "Neuro-BERT: Rethinking Masked Autoencoding for Self-Supervised Neurological Pretraining." IEEE Journal of Biomedical and Health Informatics (2024).

[5] Willett, Francis R., et al. "A high-performance speech neuroprosthesis." Nature 620.7976 (2023): 1031-1036.

[6] Metzger, Sean L., et al. "A high-performance neuroprosthesis for speech decoding and avatar control." Nature 620.7976 (2023): 1037-1046.

**Questions:**

* Are decoding words the same as decoding pronunciations in [1][2]? Is it possible that the sentiment is associated with words that the decoding task is not speech but sentiment?

**References**

[1] Willett, Francis R., et al. "A high-performance speech neuroprosthesis." Nature 620.7976 (2023): 1031-1036.

[2] Metzger, Sean L., et al. "A high-performance neuroprosthesis for speech decoding and avatar control." Nature 620.7976 (2023): 1037-1046.

**Limitations:**

The authors discussed the drawbacks of the proposed framework which is essentially close-set prediction rather than open-set. Potential negative societal impact was not mentioned.

---

> ### Author Rebuttal · Authors · 2024-08-06
>
> We are deeply grateful to you for the thorough review and constructive comments, which have greatly assisted us in improving the quality and presentation of our manuscript. Please see below for the point-by-point responses to your comments.
>
> **Answer to W1:** It’s true that the BEiT framework is popular; it has been used widely in recent years. The core idea is that it introduces discrete tokenization for mask modeling, which helps extract semantic information from data, thus enhancing model performance. For the same reason, we adopt this idea for sEEG decoding. However, we think the novelty of our study is that we are the first to find a way to adopt this idea to sEEG data for speech decoding properly and demonstrate why and how this idea works with a solid neuroscience foundation. DeWave[R1] indeed has attempted to use VQ-VAE (but not including MAE) to enhance language decoding based on EEG. However, the performance of directly applying their method to sEEG data was relatively low; see the performance of Du-IN (vqvae+vq)(44%). Besides, due to the brain's desynchronization nature[R2] during awake tasks, only specific brain regions are related to tasks. Therefore, only after region-specific channel selection can the Du-IN VQ-VAE model successfully reconstruct the original signals, thus identifying the fine-grained state of brain regions for mask modeling. Please see Table 2 in the general rebuttal pdf for more details. Moreover, we achieve the best decoding performance, only requiring about one electrode. It is really important because it reduces the medical burden and potential risks for patients.
>
> **Answer to W2:** Thank you for the helpful suggestion. LaBraM(11%), TS-TCC(24%), and Neuro-BERT(49%) are added for comparison; please refer to Table 1 in the general rebuttal pdf for details. Since LaBraM outperforms BIOT on multiple EEG tasks, we report the performance of LaBraM instead of BIOT. We can see that BrainBERT(5.6%) & Brant(8.7%) & LaBraM(11%) all performs worse than other models. For the reason of (BrainBERT,Brant,LaBraM)'s relatively low performance, please refer to the general rebuttal for more details. TS-TCC uses 1D depthwise convolutions as ours to capture local features. TS-TCC performs worse than CNN-BiGRU due to the lack of GRU for temporal integration. Neuro-BERT outperforms EEG-Conformer since it introduces MAE to learn contextual embeddings. We have cited these works in our revised paper, which is available in the Anonymous Github Repo at line 328 in the initial manuscript.
>
> **Answer to W3:** Thank you for the helpful suggestion. We reproduce BrainBERT(5.6%) and Brant(8.7%) strictly following their original implementation.  We have uploaded our reproduction to the anonymous repository, which is available in the Anonymous Github Repo at line 328 in the initial manuscript. We also uploaded their original codes and checkpoints. You can directly use their checkpoints to check whether the reproduction is the same; please see README.md in the repository for more details. Brant will load the Brant-Large checkpoints provided by the original Brant paper. In our implementation, we use the Brant-Tiny model for pre-training and evaluation. The length of the patch segment is changed from 6 seconds to 1 second for a fair comparison with LaBraM.
>
> **Answer to W4:** Thank you for the helpful suggestion. There are indeed some studies on open vocabulary, but the problem is that all of these studies are based on ECoG or microelectrode arrays. These technologies may cause more brain damage than sEEG technology in clinical practice. More importantly, we found that only one electrode was needed for decoding, further reducing clinical risks. At the same time, unlike the other two technologies that can only obtain signals from the cortex, sEEG electrodes are implanted inside the brain. Due to the existence of cortical columns, sEEG can obtain more accurate and hierarchical information. The sEEG signal is fundamentally different from other technologies’, and there was lack of methodology to draw on in this case, thus decoding the limited vocabulary from sEEG signals was still a relatively new problem.
>
> **Answer to Q1:** Thank you for the valuable feedback. Yes, decoding words and decoding pronunciations are essentially the same in that both are decoding speech-motor signals. And we are not sure what you mean by “sentiment”. If you do mean sentiment(emotional information), then we think this is unlikely to affect decoding because almost all of the words we chose were neutral words and did not contain any sentimental information. But if you mean semantics, we cannot rule out the involvement of semantic information in decoding. Because although our task setting is to collect speech-motor information, it is very likely that humans also perform corresponding semantic processing when reading words. This is indeed an interesting question, but it requires more elaborate experimental settings and corresponding data to verify this.
>
> **Reference:**
>
> [R1]Duan Y, Zhou J, Wang Z, et al. Dewave: Discrete eeg waves encoding for brain dynamics to text translation[J]. arXiv preprint arXiv:2309.14030, 2023.
>
> [R2]Buzsaki G. Rhythms of the Brain[M]. Oxford university press, 2006.

---

> > ### Comment · Reviewer_t1mV · 2024-08-12
> >
> > I thank the authors for their response and the additional experiments conducted within the limited rebuttal time window. I noticed that some of the concerns I raised are shared by other reviewers. Specifically, I still find the following concerns unaddressed and believe that this manuscript needs significant revisions before it can be accepted.
> >
> >
> > * Limited Novelty from the Algorithmic Perspective: Although the authors claim to be the first to migrate the BEiT framework for sEEG decoding, I find the novelty of this work to be largely limited. After carefully comparing the proposed Du-IN with LaBraM, I observed a more serious issue. **Figure 2 in Du-IN is exactly identical to Figure 1 in LaBraM, with spatial embedding removed. Figure 3 in Du-IN is exactly identical to Figure 2 in LaBraM, with spatial embedding removed. Furthermore, the modified attention mechanism described in Equation 3 of Du-IN is exactly identical to Equation 3 of LaBraM.**  This suggests that the proposed Du-IN may simply be a direct adaptation of LaBraM with spatial embedding removed. The authors are strongly recommended to explain this issue.
> >
> > * Regarding unfair comparisons. Given the astonishing similarities between the proposed Du-IN and LaBraM, it is surprising that the authors report approximately 60% accuracy for Du-IN and only around 10% accuracy for LaBraM. Although the authors claim that "all baselines are reproduced strictly following their original code," the baselines should be optimized for differences in decoding tasks and underlying data modality. The authors mention that the baselines were "optimized," but no details are provided on how these optimizations were performed. More transparency is needed in this regard.

---

> ### Author Response · Authors · 2024-08-12
> **Response to Reviewer t1mV**
>
> We truly appreciate your effort in helping us to strengthen our study. Please see below for the point-by-point responses to your comments.
>
> **Answer to Point 1:** Regarding the novelty of our approach, we have already provided a comprehensive response to you and Reviewer eGF6. We disagree with your statement, “the proposed Du-IN may simply be a direct adaptation of LaBraM with spatial embedding removed.” The key distinction between our Du-IN and LaBraM lies in **the basic modeling unit (i.e., channel-level embedding or region-level embedding)**; please see lines 35 & 322 in our initial manuscript for more details.
>
> LaBraM is designed as an EEG foundation model, which embeds different channels into channel-level embeddings separately. In contrast, our Du-IN uses depthwise convolution to fuse multiple channels into region-level embeddings. LaBraM asserts that their EEG foundation model, with this specific embedding design, can be applied to any EEG decoding task. However, as we show in Table 2 in the general rebuttal pdf, sEEG and EEG signals are fundamentally different. Besides, we found this specific architecture design may not be effective enough for some cognition tasks, e.g., speech decoding, which demands intricate processing in specific brain regions. We believe the core issues with LaBraM-like designs (including Brant and LaBraM) are:
>  - Overlook the fact that the same pattern in different brain locations may have different meanings, although they attempt to use spatial embeddings to identify different channels.
>  - Don’t encourage region-level embeddings either from architecture or loss.
>
> In general, our proposed Du-IN is essentially different from LaBraM except for the VQ-VAE+MAE pre-training framework (which is widely used in the field of representation learning). The VQ-VAEs in our proposed Du-IN and LaBraM are extracting tokens from totally different levels (i.e., channel-level or region-level). Besides, our reconstruction objects are totally different. While the pre-training figure may appear similar, we just used the same visual style. Regarding the attention mechanism, it was originally proposed in [R1] and has been widely adopted by many studies[R2-4].
>
> **Answer to Point 2:** For the statement “the astonishing similarities between the proposed Du-IN and LaBraM,” we have clarified this in the above response. The significant difference in performance largely stems from them. As shown in Table 1 of the general rebuttal pdf, all models with region-level embeddings outperform the current foundation models that don’t encourage region-level embeddings. We should note that the similar VQ-VAE+MAE pre-training framework doesn't affect one of the core claims of our study -- **the specific design of current foundation models may not be effective enough for some cognition tasks, e.g., speech decoding.**
>
> For the statement “the baselines should be optimized for differences in decoding tasks and underlying data modality,” we have already optimized the hyperparameters of baseline models. The model details (including the modified hyperparameters and which values have been explored) have already been added to the revised paper in the Anonymous Github Repo at line 328 in the initial manuscript. These concerns have already been addressed by Reviewer bbQK.
>
> Besides, we should note that **the core designs of current foundation models (e.g., Brant, LaBraM)** include:
>  - sharing the embedding block across different channels,
>  - modeling spatial relationships with a Transformer, either with or without spatial embeddings,
>
> These elements are what qualify them as foundation models. If you're suggesting we modify these designs, they would no longer function as foundation models:
>  - they would not be suitable for pre-training across multiple subjects,
>  - they would lose the benefits of the foundation models' scaling law -- “the more data there is, the better the performance.”
>
> **Overall comment:** The novelty of our study still holds, i.e., we are the first to find a way to **properly** adopt this idea to sEEG data for speech decoding and demonstrate why and how this idea works with a solid neuroscience foundation.
>
> **References:**
>
> [R1]Dehghani M, Djolonga J, Mustafa B, et al. Scaling vision transformers to 22 billion parameters[C]//International Conference on Machine Learning. PMLR, 2023: 7480-7512.
>
> [R2]Muennighoff N, Rush A, Barak B, et al. Scaling data-constrained language models[J]. Advances in Neural Information Processing Systems, 2024, 36.
>
> [R3]Tu T, Azizi S, Driess D, et al. Towards generalist biomedical AI[J]. NEJM AI, 2024, 1(3): AIoa2300138.
>
> [R4]Driess D, Xia F, Sajjadi M S M, et al. Palm-e: An embodied multimodal language model[J]. arXiv preprint arXiv:2303.03378, 2023.

---

### Official Review · Reviewer_gwJp · 2024-07-10

**Soundness:** 4
**Presentation:** 3
**Contribution:** 3
**Rating:** 8
**Confidence:** 5

**Summary:**

SETTING: Decoding spoken speech from invasive neural recordings, to wit, sEEG.  In particular, the MS aims to classify 61 independently produced (Chinese) words.

APPROACH: The focus of this study is on unsupervised pretraining via autoencoding, with either (a) vector-quantization or (b) masking.  Masks are 100-ms long and cover all channels.  For both VQ and masking, the core model is a "spatial encoder" (linear + conv) followed by a (temporal) transformer.  There are ~15 hours of data per subject.  After pre-training, this model is trained end-to-end (on ~3 hours of labeled data) on the classification task.

RESULTS: The architecture outperforms other architectures proposed for this task, and masking does appear to improve performance.

**Strengths:**

This is a nice study and the results are compelling.  Temporal masking looks promising as a generic technique for sEEG data.  The dataset is extremely useful and is likely to become the basis of benchmarks for future research in this area.

**Weaknesses:**

The paper does not address some important questions (admittedly there is not much space), which I list below.

In addition to these, I list here some minor issues:

--For comparing with other results in the decoding literature, it would be very useful to report cross entropy in addition to (or instead of) accuracy, since the latter can only be interpreted in light of the fact that there are 61 classes.

--The reader shouldn't have to go to Appendix C.3 just to find out what the names in Table 2 mean.  Also, what does "CLS" stand for?

--The (vqvae+vq) and (vavae) rows in Table 2 should be marked as "pre-trained."

--The MS calls the token's temporal width the "perception field."  Is this a standard term?  Perhaps the authors mean "receptive field"?

--The authors use "codex" where they mean to use "code."

**Questions:**

(1) The spatial encoder:  The name suggests that its convolutions are across channels, but this is inappropriate across sEEG electrodes (and within electrodes should take into account the spatial layout of channels).  Can the authors clarify this?  (I see in Appendix H that the *linear layer* combines channels.)

(2) The authors give some reasons for the superiority of their model over BrainBERT and Brant, but it would be helpful to have some insight on the performance relative to other models as well.  E.g.,

(a) The MS focuses on pretraining, but in fact it appears that the biggest gain over the baselines comes from the architecture itself (Du-IN trained fully supervised)---an improvement of ~25 percentage points over the CNN-BiGRU!  Can the authors shed light on this difference?

(b) The EEG-conformer likewise uses a "temporal transformer," but performance is ~10 percentage points lower than the supervised-trained Du-IN.  What do the authors see as the key difference here?

(c) The authors mention LaBraM but don't compare Du-IN to it.  It would be nice to see this comparison as well....

(3) Intracranial EEG data is generally (but not always!) filtered into the range of ~75-150, because otherwise the low frequencies, which have higher power, tend to dominate.  (The low frequencies can be provided as separate channels.)  This is a problem because low frequenceies generally contain much less information about speech.  In this study, frequencies below 70-Hz are not filtered out.  Indeed, it appears that reconstructions under vector quantizations retain mostly low-frequency information (Fig. 7).  Can the authors explain these results?

(4) Channel selection:

(a) "To streamline, we utilize these top 10 channels for both pre-training and downstream evaluation."  Does that mean that, for each subject, the best 10 electrodes (for that subject) were used?

(b) If the choice to use just 10 channels was made on the basis of decoding results on the test set, then the results are biased upwards.  Can the authors confirm that this choice was made based on validation data?  (Is Fig. 4b on validation data?)

(c) How does channel selection interact with model choice?  Were all models in Table 2 evaluated with just 10 channels?  Does their performance as a function of #channels also resemble Fig. 4b?

(5) It seems quite possible from Fig. 4a that the authors are decoding speech perception as much or more than speech production, since many of the most highly contributing electrodes are below the Sylvian fissure.  This would limit the use of this decoder as a speech prosthesis for people who cannot speak.  Can the authors comment on this?

On this head, although Fig. 4 is a nice summary, but it would be very useful to have a list/table of the electrodes used for decoding for each subject, labeled by cortical area.

---

> ### Author Rebuttal · Authors · 2024-08-06
>
> We are deeply grateful to you for the thorough review and constructive comments, which have greatly assisted us in improving the quality and presentation of our manuscript. Please see below for the point-by-point responses to your comments.
>
> **Answer to Q1:** Thank you for the helpful suggestion. We use 1D depthwise convolution, which convolves over time and integrates channels with learnable weights. Linear layer is fully-connected layer, which maps sEEG signals with 10 channels to 16 dimensions. We have clarified these in our revised paper, which is available in the Anonymous Github Repo at line 328 in the initial manuscript.
>
> **Answer to Q2:** Thank you for the valuable feedback. Architecture itself indeed matters.
>
> 1. EEG-Conformer(45%) outperforms CNN-BiGRU(32%) is because Transformer can better integrate global temporal information compared to RNN-based model.
>
> 2. Du-IN(56%) outperforms EEG-Conformer(45%) is because:
>    - The difference between our depthwise convolution and their TS-convolution. Depthwise convolution applies different convolution kernels to different channels, whereas TS-convolution applies the same convolution kernel to different channels. Brain signals are different from other time series, e.g., traffic. The same pattern in different brain locations may have different meanings. Depthwise convolution can better spatially integrate brain channels to produce meaningful tokens.
>    - Our carefully designed convolution kernels for sEEG data. The first convolution layer covers about 20ms window on the brain signal with 1000Hz sampling rate, which can capture sufficiently fine-grained features to facilitate the following integration of low-frequency and high-frequency features for speech decoding.
>
> 3. Thank you for the helpful suggestion. LaBraM(11.53%) and LaBraM-PopT(11.78%) are added for comparison; please refer to Table 1 in the general rebuttal pdf for details. For the reason for LaBraM’s relatively low performance, please refer to the general rebuttal for details.
>
> **Answer to Q3:** Thank you for the valuable feedback.
>
> 1. This is indeed an important point. Specifically, although many previous studies believe that 75~150Hz (high gamma) band signals dominate speech information, this is because high gamma band signals are believed to mainly reflect the spike information of neurons[R1]. However, in recent years, more neuroscience studies have found that LFP can also decode neural information well[R2-R5], so we retained data from all frequency bands, giving us better performance.
>
> 2. It's true that the reconstructed signal is mainly low-frequency information. Our method may not be the best to identify fine-grained states of specific brain regions since previous studies[R2-R5] found that all frequency bands contribute to speech decoding. In the future, it will be interesting to see whether introducing the STFT loss proposed in [R6], which further balances the reconstruction of low and high frequencies, leads to better performance.
>
> **Answer to Q4:** Thank you for the valuable feedback. (a) Yes, we selected the best 10 channels for each subject. (b) In the channel selection stage, we use the last training checkpoint for selection without referring to [valid,test]-set. So, these channels are selected purely based on train-set. We also tried to use the accuracy of valid-set to select channels, and the channels selected were the same. (c) Yes, all models are evaluated using the same 10 channels, and the performance of other models as a function of #channels is similar to Fig. 4b.
>
> **Answer to Q5:** Thank you for the helpful suggestion. Due to the difference in the number of electrodes on the left and right brain, we standardized and rendered them separately when drawing the map. We have changed to joint standardization and rendering. In fact, the important site is still the language motor area of the left brain. We have made this table and added it to the article, but due to the word limit of the rebuttal, we only show one subject here. Please see our revised paper for more details.
>
> |Subjs|MNI|coord-|-inate|Brain Region|ACC(%)|
> |:--|:--:|:--:|:--:|:--:|:--:|
> ||__x__|__y__|__z__|||
> ||-57|-16|19|Central Opercular Cortex L||
> ||-61|-16|20|Postcentral Gyrus L||
> ||-54|-16|17|Central Opercular Cortex L||
> ||-67|-15|23|Postcentral Gyrus L||
> |01|-25|-30|49|White L|71.25|
> ||-64|-15|21|Postcentral Gyrus L||
> ||-51|-17|16|Central Opercular Cortex L||
> ||-22|-31|49|White L||
> ||-48|-17|15|Central Opercular Cortex L||
> ||-18|-32|49|White L||
>
> **Answer to W1&2&3&4&5:** Thank you for the helpful suggestion. We have made these clear in our revised paper. Due to the limited space for comments and rebuttal pdf, we provide cross-entropy results in Appendix J of our revised paper.
>
> **Reference:**
>
> [R1]Buzsáki, G., & Wang, X. J. (2012). Mechanisms of gamma oscillations. Annual review of neuroscience, 35(1), 203-225.
>
> [R2]Proix, T., Delgado Saa, J., Christen, A., Martin, S., Pasley, B. N., Knight, R. T., ... & Giraud, A. L. (2022). Imagined speech can be decoded from low-and cross-frequency intracranial EEG features. Nature communications, 13(1), 48.
>
> [R3]Ahmadi, N., Constandinou, T. G., & Bouganis, C. S. (2021). Impact of referencing scheme on decoding performance of LFP-based brain-machine interface. Journal of Neural Engineering, 18(1), 016028.
>
> [R4]Stavisky, S. D., Kao, J. C., Nuyujukian, P., Ryu, S. I., & Shenoy, K. V. (2015). A high performing brain–machine interface driven by low-frequency local field potentials alone and together with spikes. Journal of neural engineering, 12(3), 036009.
>
> [R5]Flint, R. D., Lindberg, E. W., Jordan, L. R., Miller, L. E., & Slutzky, M. W. (2012). Accurate decoding of reaching movements from field potentials in the absence of spikes. Journal of neural engineering, 9(4), 046006.
>
> [R6]Dhariwal, P., Jun, H., Payne, C., Kim, J. W., Radford, A., & Sutskever, I. (2020). Jukebox: A generative model for music. arXiv preprint arXiv:2005.00341.

---

> > ### Comment · Reviewer_gwJp · 2024-08-13
> >
> > Thanks, this is helpful.
> >
> > I have read the other, less favorable reviews, as well, but I retain my high score.  I find the comparisons to other models compelling and consistent with the results reported in those papers.  (E.g., the results in the original BrainBERT paper are for *binary classification*.  BrainBERT learns a single model for all channels! so unsurprisingly doesn't work well on difficult tasks.  The model from Moses et al. achieved ~38% accuracy on a 50-word task with ~3 hours of training data [Fig. S8 in that work], which is very close to what the authors report [32% accuracy on a 61-word task, ~3 hours of training data].  Etc.)

---

> > > ### Author Response · Authors · 2024-08-13
> > > **Response to Reviewer gwJp**
> > >
> > > We sincerely appreciate your effort in helping us enhance the paper and the positive recognition of our work.

---

### Official Review · Reviewer_1qa8 · 2024-07-11

**Soundness:** 4
**Presentation:** 4
**Contribution:** 4
**Rating:** 7
**Confidence:** 3

**Summary:**

The authors introduce Du-IN, a novel decoding technique for speech production the leverages sEEG data. To achieve their goals, the authors gathers a novel dataset. The Du-IN model learns multi-variate (across channel) representations from sEEG signal using a discrete cookbook and a masked modeling based self-supervised training. The proposed algorithm achieves SoTA performances on the speech 61-words classification task. Furthermore, analysis of optimal model configuration and ablation studies reveal that best performances are achieved with electrodes in brain regions known to be involved in speech production.

**Strengths:**

The paper presents strong results of high significance for the field of speech decoding. In particular:
- The authors have collected a novel sEEG dataset, which will be made publicly available, hence fostering further progress in the field of brain signal decoding.
- The authors propose a novel framework for sEEG signal decoding - the Du-IN algorithm, which achieves SoTA performances, thus significantly advancing the field of brain decoding.
- Authors provide additional ablations studies to validate design choices and recover known results from Neuroscience, strengthening the overall soundness of their technique.
- The paper is well written and clearly organized, with a detailed Appendix providing further details.

**Weaknesses:**

- In Table 2 both the Du-IN (vqvae+vq) and Du-IN (vqvae) are marked as not pre-trained. However in the Appendix C.3 it is stated that their Neural Encoder weights are loaded from the pre-trained Du-IN VQ-VAE model.
- Minor spelling: Line 156 DuIR $\to$ DuIN

**Questions:**

- There seems to exist substantial variation of decoding accuracy across subjects. Is this difference due to different electrode locations, or number of electrodes (even if only 10 channels seem to be optimal) or are other concerns with data quality relevant?
- Could the manual selection of channel be partially subsumed by a sparsity regularization term on the channel-to-embedding projection matrix?

**Limitations:**

The authors have adequately addressed the limitation of their work with a dedicated section in the paper.

---

> ### Author Rebuttal · Authors · 2024-08-06
>
> We are deeply grateful to you for the thorough review and constructive comments, which have greatly assisted us in improving the quality and presentation of our manuscript. Please see below for the point-by-point responses to your comments.
>
> **Answer to W1 & W2:** Thank you for the valuable feedback. These have been corrected in the revised paper, which can be accessed in the Anonymous Github Repository at line 328 in the initial manuscript.
>
> **Answer to Q1:** This difference is mainly due to different electrode locations, as we find that decoding vocal production is mainly related to specific brain regions (i.e., vSMC, STG), which is consistent with the previous neuroscience findings.
>
> **Answer to Q2:** Thank you for the helpful suggestion. Yes, this modification will make the channel selection more automatic and general.

---

### Official Review · Reviewer_bbQK · 2024-07-12

**Soundness:** 2
**Presentation:** 3
**Contribution:** 2
**Rating:** 5
**Confidence:** 4

**Summary:**

The authors introduce a neural network architecture, a self-supervised learning (SSL) method for pre-training models for sEEG decoding and a novel dataset.
- The architecture starts with segmenting the sEEG signal along the time dimension, followed by a linear module combining information across the different sEEG channels into embedding vectors and ending with a transformer with attention along the time dimension.
- The SSL method has two stages:
    - 1. The architecture followed by an additional decoder network is trained to reconstruct the original signals (autoencoder). However, the output of the transformer is not directly passed to the decoder, instead, each token is replaced with the closest embedding vector from a trainable codebook (vector quantisation).
    - 2. Time windows of the input are masked and the architecture is trained to predict which codebook embedding represents best each masked window.
- The dataset introduced contains sEEG recordings of 12 subjects while they were speaking aloud Chinese words from a vocabulary of size 61.

In the experiments, the models were pre-trained using the SSL tasks using unlabelled data from a single subject and fine-tuned on labelled data from the same subject for classifying the 61 words.
The authors conducted different ablation studies regarding pre-training (no pre-training, and only step 1 of pre-training using quantized features for fine-tuning or not), dataset, codex size and dimensionality, and length of the time windows.
They also compare with pre-existing methods, two including a pre-trining phase and two without. The two with pre-training are pre-trained on all 12 subjects.

**Strengths:**

*Significance*: As pointed out by the authors, sEEG speech datasets are rare. Their promise of publishing the dataset they introduce here is good news for the community as it will lower the entry threshold for future research. Additionally, they demonstrate how SSL-based pre-training  allows improving performance compared to only supervised training.

*Clarity*: The text has a good structure and is well-written. The figures also help in understanding the method.

*Originality*: SSL techniques are still rather novel to the BCI community. This work sheds light on some of them.

**Weaknesses:**

**Major**
1. The authors pre-train their method on the test subject (Lines 201-202) whereas the baseline methods with pre-training (BrainBERT and Brant) were pre-trained on all 12 subjects (lines 511-518). This does not allow for a fair comparison. Additionally, this choice is not properly justified.
2. The authors mention in section 4.5 Ablation study, that they found optimal hyperparameters for their architecture (codex of size 2048 and dimension 64, and perceptual field of 100ms). They use these hyper parameters to compare to the other methods (see Table 5). However, the manuscript does not mention the use of a holdout set for conducting this ablation study. This is problematic for two reasons:
    1. This suggests the method could have a low capacity to generalise be overfitting to their dataset
    2. This is not fair for the other methods as no hyperparameter optimisation is mentioned for them.
3. The authors mention they resampled their signals to 1000Hz (line 210). However, some architecture compared have been introduced with different sampling rates (Brant used 250Hz and EEG-Conformer used 200 and 250Hz). The text does not mention any adaptation of the architectures compared or of the sampling rate. This is problematic as a convolution kernel supposed to cover 1s of signal at 250Hz will only cover 0.25s at 1000Hz.
4. Some aspects of the method are undefined or unclear. Please see the Questions section below. These aspects need to be clarified in the manuscript.


**Minor**
1. Because of the flaws in experimental design described above, the statements about the superiority of their method compared to others (lines 14, 71, 248 and 318) are unjustified.
2. All figures should be in a vectorized format.
3. The text in the figures should be approximately constant and the same as the article’s text. It is too small in figures 5 and 7. It is on the limit in figures 1,2 and 3.
4. The term “fusing channels” is used lines 16, 39, 108 and 117 but is rather unclear. It is only explained line 134.
5. Line 29: provide examples of non-invasive recording methods (EEG, MEG…)
6. Section 2.3, additional publication the authors should be aware:
    1. A preprint of Brant-2 is now available https://arxiv.org/abs/2402.10251
    2. SSL methods trained on EEG with attention mechanism for combining information across channels: EEG2Rep (https://arxiv.org/abs/2402.17772) and S-JEPA (https://arxiv.org/abs/2403.11772).
7. Line 54: Indeed, there is a lack of open sEEG recordings but some still exist. In reference [1], they mention they use a public dataset, in [16] they mention their data is available upon request and Brant-2 (not cited here), they use a 26-subject public dataset. Existing sEEG datasets should be cited.
8. Line 98: typo “Metzger1”
9. As I understand, the terms “Du-IN Encoder”, “neural transformer”, “neural encoder”, and “neural tokeniser” all refer to the architecture in figure 2. If this is indeed the case, it would improve readability to chose one of them and stick to it throughout the manuscript.
10. Line 166: missing “to” in “$z_c$ to get”
11. Figures 2 and 3: It would improve readability to place the different variables ($e^p_i$, $e_i$, $e^t_i$, $c_j$,…) on these figures.
12.  Line 162: for consistency, you could use $j$ as index for the codex instead of $i$.
13. Lines 180-193: the objective function of the masked training is unclear (see questions).
14. Line 247: Please mention that the baseline descriptions can be found in appendix C.3.
15. Line 301: The research line of Feng et al. was already explored by Christian Herff in 2015 (https://doi.org/10.3389/fnins.2015.00217)  and is still today (https://www.sciencedirect.com/science/article/pii/S1053811923000617). It could be interesting to draw a parallel.



**Overall comment**
The introduction of a novel sEEG dataset is valuable but the evaluation of the machine learning algorithms that are introduced is poor. Therefore,  I believe NeurIPS might not be the best fit for this publication.

**Questions:**

1. Lines 134-140: The description of the spatial encoder is ambiguous:
    1. Does “linear” mean fully-connected layer? Appendix C.1 mentions “Linear Projection: 10->16” which would mean that an operation is applied to each channel independently and that the layer is actually not a fully-connected one.
    2. Convolutions are mentioned, but over which dimensions? Over time? Channels? Both?
2. Figure 3: What are the numbers below “l2-norm” and bellow “neural regressor” corresponding to?
3. Why did you use a different signal length for pre-training and for downstream classification?
4. Line 210: Why did you use a sampling rate of 1000Hz? With a low-pass filtering at 200Hz, a sampling rate of 400Hz would be sufficient
5. Line 211: Did you apply the normalisation on each channel independently (i.e., mean and std computed for one channel only) or using all channels together?
6. Line 222: How is the training time split between VQ-VAE and MAE training?
7. Line 228, “selected from the validation set”: does this mean early stopping? Was it performed using the validation loss, validation accuracy or another metric?
8. Table 2, “Du-IN (mae)”: this row corresponds to the scores obtained after a VQ-VAE training followed by a MAE training. Is the first step exactly “vqvae” or “vqvae+vq”? To lift the ambiguity, you could rename the row to “Du-IN (mae+vqvae)” or “Du-IN (mae+vqvae+vq)”.
9. Why refer to your method as a “VAE”? As I understand, for a given input, the “reconstructed sEEG signals” are obtained in a deterministic manner (please, correct me if I’m wrong am wrong). The term “variational” in VAE suggests there is a stochastic operation involved.
10. Lines 180-193: the objective function of the masked training is unclear:
    1.  Line 188: “neural tokens” is ambiguous. What is being predicted?
    2. Equation (7) and (8): how are the $z_i$ obtained?
    3. Equation (8): $m_i$ is undefined. What is the meaning of $m_i$?

**Limitations:**

The authors included a Limitations section in their manuscript in which they already mention other approaches to speech decoding, at the phonemes level and at a semantic level.

It could be relevant to remind the reader that sEEG recordings are invasive. Therefore, they are only done with patients who would require them anyway for medical reasons (epilepsy monitoring, etc…). And this presents a major obstacle to making these methods widely accessible and usable.

---

> ### Author Rebuttal · Authors · 2024-08-06
>
> We are deeply grateful to you for the thorough review and constructive comments, which have greatly assisted us in improving the quality and presentation of our manuscript. Please see below for the point-by-point responses to your comments.
>
> **Answer to Major W1:** Thank you for the helpful suggestion. We have added DuIN(poms)(59.18%), which is pre-trained on all subjects for a fair comparison. Since BrainBERT and Brant support pre-training on multi-subjects, this operation can maximize the downstream performance according to their reported scaling law.
>
> **Answer to Major W2:** Thank you for the valuable feedback.
>
> 1. All experiments are conducted on the same machine with the same set of random seeds; the train/valid/test splits are the same across different models.
>
> 2. In fact, the performances of BrainBERT(7.5%) and Brant(12.4%) we report are the results after model optimization, and their performance with the original hyperparameter settings is even lower(5.6%&8.7%). Please see the general rebuttal for more details. We have clarified these in our revised paper, which is available in the Anonymous Github Repo at line 328 in the initial manuscript.
>
> **Answer to Major W3:** Actually, we resampled the signals to the required frequency of each model. Please refer to line 508 in the Appendix.
>
> **Answer to Minor W2&3&4&5&8&9&10&11&12&14:** Thank you for the valuable feedback. We have corrected all formatting, spelling, references, and wording issues. We have updated the paper in the Anonymous Github Repo at line 328 in the initial manuscript.
>
> **Answer to Minor W6&7&15:** Thank you for the helpful suggestion. We have cited these works and updated the paper in the Anonymous Github Repo at line 328 in the initial manuscript.
>
> **Answer to Q1:** Thank you for the valuable feedback.
>
> 1. Yes, we use “linear” to refer to the fully connected layer, which projects 10 channels to 16 dimensions.
>
> 2. We use `torch.nn.Conv1d` with `groups=1`, i.e., depthwise convolution over time.
>
> **Answer to Q2:** It corresponds to the index of discrete codes in the codex.
>
> **Answer to Q3:** Thank you for the valuable feedback. We followed LaBraM's settings and used 4s signals for pre-training. The signal length of downstream tasks is usually shorter than the pre-training length.
>
> **Answer to Q4:** Thank you for the valuable feedback. We tried the effects of different sampling rates and finally found that there was not much difference between 400 and 1000Hz, but 1000Hz is slightly better. At the same time, we use a sampling rate of 1000Hz to minimize the difference with the BrainBERT model because they also perform sEEG language-related tasks and use 2000Hz.
>
> **Answer to Q5:** We did it on each channel independently; please refer to line 211 in Section 4.2.
>
> **Answer to Q6:** The training time splits are the same.
>
> **Answer to Q7:** Thank you for the valuable feedback. Yes, this means early stopping, and we performed using the validation accuracy.
>
> **Answer to Q8:** Du-IN (mae) is obtained after a VQ-VAE training followed by a MAE training, i.e., loading weights from the Du-IN MAE. “+vq” means we use quantized embeddings after the vector-quantizer for the downstream task, which is a specific operation when loading weights from the Du-IN VQ-VAE.
>
> **Answer to Q9:** Thank you for the valuable feedback. It's true that our method is not variational; we followed the name “VQ-VAE” proposed in [R1].
>
> **Answer to Q10:** Thank you for the valuable feedback.
>
> 1. The “neural tokens” means the code index in the codex of the Du-IN VQ-VAE model.
>
> 2. It is obtained from Equation (4).
>
> 3. Sorry for the confusion, it is the $\delta(i\in\mathcal{M})$ from Equation (6).
>
> **References:**
>
> [R1]Aaron Van Den Oord, Oriol Vinyals, et al. Neural discrete representation learning. Advances in 434 neural information processing systems, 30, 2017.

---

> > ### Comment · Reviewer_bbQK · 2024-08-09
> >
> > Thank you for your detailed rebuttal and answering all my points. Nevertheless, I still have three concerns:
> > - **Major W1** Thank you for adding the DuIN(poms) pipeline. However, as your experiment pointed out, pre-training on multiple subjects does not necessarily lead to a better downstream performance. Indeed, DuIN(poms) is 3% bellow Du-IN(mae) trained within-subject only. Therefore, I believe it would be fair to report the within-subject pre-training performance for all the baselines.
> > - **Major W2** Thank you for adding details regarding the hyperparameters optimisation of the baselines. However:
> >     - In your answer, you mention that “the performances of BrainBERT(7.5%) and Brant(12.4%) we report are the results after model optimization”, but your revised article says the opposite lines 584 and 590. Which one is correct?
> >     - You now mention that the hyperparameters of some of the baselines are optimized but not which hyperparameters are optimized. For reproducibility, you should list all the hyperparameters optimized and detail which values have been explored.
> > - **Q9** If the naming scheme in previous publications is misleading to the reader, I suggest not following it.

---

> ### Author Response · Authors · 2024-08-10
> **Response to Reviewer bbQK**
>
> We wholeheartedly appreciate and deeply cherish your efforts to help us strengthen our manuscript. Please see below for the point-by-point responses to your comments.
>
> **Answer to Major W1:** Thank you for the helpful suggestion. The performances of (BrainBERT(5.33%), Brant(7.32%), BrainBERT$\oplus$(6.72%), Brant$\oplus$(11.16%)) pre-trained within each subject are added to the revised paper; please see Table 2 for more details. All checkpoints are uploaded to the Anonymous Github Repo. Their performances are slightly lower than models pre-trained on all subjects, which aligns with their reported scaling law (the core property of the sEEG foundation model) -- “the more data there is, the better the performance.” Besides, this result doesn't undermine the core claim of our study -- current sEEG foundation model may not be the best fit for more difficult tasks, e.g., speech decoding, which demands intricate processing in specific brain regions.
>
> **Answer to Major W2:**
>
> 1. Thank you for the valuable feedback. The hyper-parameter optimized versions of BrainBERT and Brant are called BrainBERT$\oplus$ and Brant$\oplus$, respectively. Their detailed descriptions are added in Appendix B in our revised paper.
>
> 2. Thank you for the helpful suggestion. Model details about baseline models matter! The details about optimized hyper-parameters and the details about which values have been explored are added in Appendix B in our revised paper.
> Providing details about which values have been explored offers valuable insights, although some previous works[R1] only provided the final version of optimized hyperparameters. However, more previous works[R2-5] don’t even mention the hyperparameter optimization of baseline models. The reason why we optimize baseline models is because we want to ensure the robustness of our carefully designed model, i.e., Du-IN (w/o pre-training). Based on this, we can further evaluate the effectiveness of discrete-codex-guided mask modeling, providing both **a strong backbone framework for sEEG speech decoding** and **a potentially general self-supervision framework for sEEG decoding**.
> Additionally, we should note that the hyper-parameter optimization doesn’t affect the two core claims in our study:
>    - The superiority of introducing VQ-VAE (for extracting fine-grained brain region states) before MAE mask modeling.
>    - Current sEEG foundation model may not be the best fit for more difficult tasks, e.g., speech decoding, which demands intricate processing in specific brain regions.
>
>     Our ablation studies on the effectiveness of vector-quantized neural signal prediction (Table 4 in the general rebuttal PDF) have supported our first core claim, as both BrainBERT and Brant introduce MAE mask modeling. Furthermore, as we explained in the general rebuttal, the current sEEG foundation models (BrainBERT and Brant) show relatively low performance primarily because they don’t encourage region-level embeddings either from architecture or loss. These results demonstrate our second core claim.
>
> **Answer to Q9:** Thank you for the valuable feedback. It’s true that the name “VQ-VAE” may be misleading, but it’s widely used in the field of representation learning and its applications[R6-9]. It seems that the “VQ-VAE” has become the proper name of this architecture. To prevent any confusion, we've added additional information in our revised paper when "Du-IN VQ-VAE" is first mentioned.
>
> **References:**
>
> [R1]Yang C, Westover M, Sun J. Biot: Biosignal transformer for cross-data learning in the wild[J]. Advances in Neural Information Processing Systems, 2024, 36.
>
> [R2]Baevski A, Zhou Y, Mohamed A, et al. wav2vec 2.0: A framework for self-supervised learning of speech representations[J]. Advances in neural information processing systems, 2020, 33: 12449-12460.
>
> [R3]Jiang W B, Zhao L M, Lu B L. Large brain model for learning generic representations with tremendous EEG data in BCI[J]. arXiv preprint arXiv:2405.18765, 2024.
>
> [R4]Zhang D, Yuan Z, Yang Y, et al. Brant: Foundation model for intracranial neural signal[J]. Advances in Neural Information Processing Systems, 2024, 36.
>
> [R5]Duan Y, Chau C, Wang Z, et al. Dewave: Discrete encoding of eeg waves for eeg to text translation[J]. Advances in Neural Information Processing Systems, 2024, 36.
>
> [R6]Razavi A, Van den Oord A, Vinyals O. Generating diverse high-fidelity images with vq-vae-2[J]. Advances in neural information processing systems, 2019, 32.
>
> [R7]Dhariwal P, Jun H, Payne C, et al. Jukebox: A generative model for music[J]. arXiv preprint arXiv:2005.00341, 2020.
>
> [R8]Li S, Wang Z, Liu Z, et al. VQDNA: Unleashing the Power of Vector Quantization for Multi-Species Genomic Sequence Modeling[J]. arXiv preprint arXiv:2405.10812, 2024.
>
> [R9]Jiang B, Chen X, Liu W, et al. Motiongpt: Human motion as a foreign language[J]. Advances in Neural Information Processing Systems, 2024, 36.

---

> > ### Comment · Reviewer_bbQK · 2024-08-12
> >
> > - **W1** Indeed, I overlooked the $\oplus$ sign.
> > - **W2** Thank you for these additional details.
> > - **Q9** Ok

---

> > > ### Author Response · Authors · 2024-08-12
> > > **Response to Reviewer bbQK**
> > >
> > > We sincerely appreciate your effort in reviewing our manuscript and offering valuable suggestions to help us improve this work during the rebuttal stage!
> > >
> > > We have made complements and explanations for this work with the help of all the reviews. We would be grateful if you could confirm whether the rebuttal meets your expectations and if there is any other suggestion.
> > >
> > > If we have addressed all of your concerns, would you be willing to reconsider your rating for this work?
> > >
> > > Best regards.

---

### Official Review · Reviewer_eGF6 · 2024-07-22

**Soundness:** 2
**Presentation:** 3
**Contribution:** 2
**Rating:** 4
**Confidence:** 5

**Summary:**

Due to the locality and specificity of brain computation, existing models that learn representations of a single channel yield unsatisfactory performances on more difficult tasks like speech decoding. This paper tries to build a stereoElectroEncephaloGraphy (sEEG) foundation model with Vector Quantization (VQ) pre-training. We hypothesize that building multi-variate representations within certain brain regions can better capture the specific neural processing. To this end, we collect a well-annotated Chinese word-reading sEEG dataset, targeting language-related brain networks over 12 subjects, and we developed the Du-IN model that can extract contextual embeddings from specific brain regions through discrete codebook-guided mask modeling. Extensive comparison and ablation experiments verify that our model achieves SOTA performance on the downstream 61-word classification task.

**Strengths:**

**(S1)** This paper provides an interesting solution for the sEEG learning framework that models multiple sEEG channels by a spatial encoder and quantizes sEEG signals from the temporal aspect. To evaluate the proposed Du-IN framework, the authors collect and contribute a Chinese word-reading sEEG dataset for speech decoding (as word classification).

**(S2)** The manuscript is clearly structured and easy to follow. Empirical analysis and ablations of proposed modules provide comprehensive understandings.

**Weaknesses:**

**(W1) The employed techniques in the proposed framework lack novelty to some extent.** Firstly, except for the spatiotemporal modeling architecture for sEEG, the VQVAE tokenization and masked code modeling pre-training framework is adopted from popular BeiT frameworks [1, 2], which has been adopted to various scenarios like [3, 4, 5]. Some specially designed modules are necessary for introducing the BeiT-like framework, and the current version is not effective and well-designed from my aspect (view W2 for concerns). Secondly, the authors regard the speech decoding tasks as word classification, which might be too coarse to decode the actual meanings of human speeches. Therefore, the conducted decoding task might not be sufficient to verify the effectiveness of the Du-IN framework.

**(W2) Weak comparison experiments.** Firstly, the authors only provide a few supervised learning and self-supervised pre-training baselines in Table 2 with a single-word classification task (as speech decoding). More relevant contrastive learning [6] and masked modeling pre-training methods [7] for sEEG should be considered, which should also be referred to in the background section. Meanwhile, some supervised baselines with more parameters can be added for a fair comparison as Du-IN models. Secondly, the results of some baseline methods are very bad (e.g., BrainBERT and Brant). It’s hard to believe that pre-training models will yield worse results than random initialized supervised learning models. I suggest the authors conduct a fair and comprehensive comparison experiment.

**(W3) Not explicitly modeling multiple subjects.** Since the invasive sEEG signals for different subjects have different channels, I couldn’t find an explicit modeling strategy in the proposed encoders and pre-training framework that considers fusing or selecting information from multiple subjects. There might be some challenges, like representation conflicts and missing channels of different subjects, which will decrease the overall cross-subject performances. I suggest the authors conduct such analysis of multiple subjects, e.g., cross-validation of several subjects and transfer learning of pre-trained representation across datasets of different subjects.

### Reference

[1] MAGE: Masked Generative Encoder to Unify Representation Learning and Image Synthesis. CVPR, 2023.

[2] BEIT: BERT Pre-Training of Image Transformers. ICLR, 2021.

[3] Human Pose as Compositional Tokens. CVPR, 2023.

[4] MAPE-PPI_Towards Effective and Efficient Protein-Protein Interaction Prediction via Microenvironment-Aware Protein Embedding. ICLR, 2024.

[5] VQDNA: Unleashing the Power of Vector Quantization for Multi-Species Genomic Sequence Modeling. ICML, 2024.

[6] Time-series representation learning via temporal and contextual contrasting. IJCAI, 2021.

[7] Neuro-BERT: Rethinking Masked Autoencoding for Self-Supervised Neurological Pretraining. IEEE Journal of Biomedical and Health Informatics, 2024.

==================== Post-rebuttal Feedback ====================

After considering the authors' rebuttal and other reviewers' comments, I decided to keep my score because some of my concerns were not solved directly (e.g., W1 and W3). Meanwhile, the authors have provided some useful results (e.g., more comparison experiments) to support the effectiveness of the proposed methods. Since the revision could not be uploaded, I encourage the authors to improve the manuscript further for submission to the next conference or journal.

**Questions:**

**(Q1)** For different subjects with different channels, does the spatial encoder need different parameters for each subject? And does the output embedding of each subject in the same shape?

**(Q2)** Does the authors conduct ablation studies of masked code modeling, e.g., the masking ratio and pre-training epochs? Since the proposed Du-IN is a framework with several networks and the pre-traning & fine-tuning pipeline, it’s better to analyze and clearify each module for better understanding and practical usage.

**Limitations:**

Although the authors discussed some limitations in Sec. 5, potential negative societal impact of the proposed dataset and the Du-IN framework were not clearly discussed. More details of the implementations and the collected dataset can be provided for the sake of the neuroscience community.

---

> ### Author Rebuttal · Authors · 2024-08-06
>
> We are deeply grateful to you for the thorough review and constructive comments, which have greatly assisted us in improving the quality and presentation of our manuscript. Please see below for the point-by-point responses to your comments.
>
> **Answer to W1**:
>
> 1. It’s true that the BEiT framework is popular; it has been used widely in recent years. The core idea is that it introduces discrete tokenization for mask modeling, which helps extract semantic information from data, thus enhancing model performance. For the same reason, we adopt this idea for sEEG decoding. However, we think the novelty of our study is that we are the first to find a way to adopt this idea to sEEG data for speech decoding properly and demonstrate why and how this idea works with a solid neuroscience foundation. DeWave[R1] indeed has attempted to use VQ-VAE (but not including MAE) to enhance language decoding based on EEG. However, the performance of directly applying their method to sEEG data was relatively low; see the performance of Du-IN (vqvae+vq)(44%). Besides, due to the brain's desynchronization nature[R2] during awake tasks, only specific brain regions are related to tasks. Therefore, only after region-specific channel selection can the Du-IN VQ-VAE model successfully reconstruct the original signals, thus identifying the fine-grained state of brain regions for mask modeling. Please see Table 2 in the general rebuttal pdf for more details. Moreover, we achieve the best decoding performance, only requiring about one electrode. It is really important because it reduces the medical burden and potential risks for patients.
>
> 2. It’s true that word classification may not exactly decode semantic meanings of speech. However, we want to emphasize that this study aims to develop a system that can decode speech (vocal production) from limited intracranial data, which will help patients in the future.
>
> **Answer to W2:**
>
> 1. Thank you for the helpful suggestion. TS-TCC(24%) and Neuro-BERT(49%) are added for comparison; please refer to Table 1 in the general rebuttal pdf for details. TS-TCC uses 1D depthwise convolutions as ours to capture local features. TS-TCC performs worse than CNN-BiGRU due to the lack of GRU for temporal integration. Besides, there may be some misunderstanding about Neuro-BERT. It is developed for EEG and evaluated only on single-channel EEG tasks, e.g., seizure detection. We note Neuro-BERT is also evaluated on non-EEG tasks (e.g., Hand Gesture Recognition), which use depthwise convolution to integrate multiple channels. Neuro-BERT outperforms EEG-Conformer since it introduces MAE to learn contextual embeddings. DeWave-CLS[R1] (32%) is added, which replaces our depthwise convolutions with Conformer and has more parameters. We have cited these works in our revised paper, which is available in the Anonymous Github Repo at line 328 in the initial manuscript.
>
> 2. BrainBERT and Brant indeed outperform their randomly initialized models. Brant is a great work that serves as the medical field's foundation model and achieves SOTA on many medically valuable tasks, e.g., seizure detection. But their architecture may not be suitable for some cognition tasks, e.g., speech decoding. For the reason of their relatively low performances, please refer to the general rebuttal for more details.
>
> **Answer to W3:**
>
> Thank you for the helpful suggestion. We add Du-IN (poms)(59%), which is pre-trained over 12 subjects. The representation conflicts do exist, which performs slightly worse than Du-IN (mae)(62%), which is pre-trained on that subject. But we should note that this is not the core aim of our study. We aim to find a way to leverage redundant pure recordings of one subject to build the codebook of speech-related brain regions, which can identify fine-grained region states to improve the efficiency of mask modeling, thus improving speech decoding performance.
>
> **Answer to Q1:**
>
> Thank you for the valuable feedback. We initialize different spatial encoders for different subjects; all these spatial encoders have the same hyperparameters. The output embeddings are of the same shape. The following transformer encoder and vector quantizer are shared. We have clarified these in our revised paper, which is available in the Anonymous Github Repo at line 328 in the initial manuscript.
>
> **Answer to Q2:**
>
> Thank you for the helpful suggestion. We add these ablation studies in Table 3,5,6 in the general rebuttal pdf. For ablations on pre-training epochs:
>
> 1. We use the checkpoints of Du-IN VQ-VAE according to the specified epochs to train the Du-IN MAE model for 400 epochs. Once the reconstruction loss of the Du-IN VQ-VAE model converges, the Du-IN VQ-VAE model can extract the state of the brain region well, thus leading to better performance.
>
> 2. We use the checkpoints of Du-IN MAE according to the specified epochs for downstream classification. Once the mask modeling loss of the Du-IN MAE model converges, the Du-IN MAE model learns robust contextual embeddings, thus leading to better performance.
>
> **References:**
>
> [R1]Duan Y, Zhou J, Wang Z, et al. Dewave: Discrete eeg waves encoding for brain dynamics to text translation[J]. arXiv preprint arXiv:2309.14030, 2023.
>
> [R2]Buzsaki G. Rhythms of the Brain[M]. Oxford university press, 2006.

---

> > ### Comment · Reviewer_eGF6 · 2024-08-13
> > **Feedback to Authors' Rebuttal**
> >
> > Thanks for the comprehensive responses to my concerns! I acknowledged and agreed with some points, e.g., I appreciate the additional cooperation experiments for W2 (it is better to add them to comparison tables). However, I found several essential issues not well solved and decided to keep my score at the current stage.
> >
> > * (W1) The concerns of the novelty of the proposed method and whether it could called decoding speech still existed. Despite previous works in neuroscience applications being accepted with a naive design of VQ-based methods, it does not mean more recently proposed research should follow their standards. The NeurIPS community requires innovation of the proposed methods or the research problem to some extend. Meanwhile, this work regards speech decoding as a coarse word classification task, which could be tackled well with a more complex formulation. Therefore, I suggest the authors to improve their method and resubmit to the next conference or a journal.
> >
> > * (W3) The issue of multiple subjects has been a challenging problem of sEEG-related tasks for a long time. I think the authors should take it into consideration or employ a more complex decoding formulation, as I mentioned in W1.
> >
> > * Regarding responses to W2, Q1, and Q2, I have no further concerns and suggest the authors revise the manuscript with these discussions.

---

> > > ### Author Response · Authors · 2024-08-13
> > > **Response to Reviewer eGF6**
> > >
> > > We truly appreciate your effort in helping us to strengthen our study. Please see below for the point-by-point responses to your comments.
> > >
> > > **Answer to Point 1 (W1):**
> > > 1. Regarding the novelty of our approach, we have already provided a comprehensive response previously. DeWave[R1] is a great work that attempts to introduce VQ-VAE for language decoding. However, due to the low signal-to-noise ratio of the EEG dataset[R2-3] and their flawed “teacher-forcing” evaluation method[R4], the effectiveness of their VQ-VAE application is still under exploration. From this point, we are the first to find a way to adopt this idea to sEEG data for speech decoding properly and demonstrate why and how this idea works with a solid neuroscience foundation.
> > > 2. We think that word classification still belongs to speech decoding because it is the purpose of our study, which potentially helps individuals who cannot speak regain the ability to engage in simple daily communication using a limited set of words. Moreover, the electrodes we use for decoding are primarily located in the language-motor brain regions, which are exactly responsible for speech.
> > >
> > > **Answer to Point 2 (W3):** We think that the issue of multiple subjects has been a challenging problem of sEEG-related tasks for a long time primarily due to the lack of public sEEG datasets instead of the algorithm itself[R5]. Besides, we want to emphasize that this is not the aim of our study.
> > >
> > > **Answer to Point 3:** The additional experiments and responses to W2&Q1&Q2 have already been added to the comparison tables in the general rebuttal pdf and our revised paper. Since NeurIPS 2024 does not currently allow direct manuscript revisions, only a one-page rebuttal PDF can be submitted. We have uploaded the revised paper to an Anonymous GitHub Repo; please refer to the general rebuttal for more details.
> > >
> > > **References:**
> > >
> > > [R1]Duan Y, Zhou J, Wang Z, et al. Dewave: Discrete eeg waves encoding for brain dynamics to text translation[J]. arXiv preprint arXiv:2309.14030, 2023.
> > >
> > > [R2]Hollenstein N, Rotsztejn J, Troendle M, et al. ZuCo, a simultaneous EEG and eye-tracking resource for natural sentence reading[J]. Scientific data, 2018, 5(1): 1-13.
> > >
> > > [R3]Hollenstein N, Troendle M, Zhang C, et al. ZuCo 2.0: A dataset of physiological recordings during natural reading and annotation[J]. arXiv preprint arXiv:1912.00903, 2019.
> > >
> > > [R4]Jo H, Yang Y, Han J, et al. Are EEG-to-Text Models Working?[J]. arXiv preprint arXiv:2405.06459, 2024.
> > >
> > > [R5]Chen J, Chen X, Wang R, et al. Subject-Agnostic Transformer-Based Neural Speech Decoding from Surface and Depth Electrode Signals[J]. bioRxiv, 2024.

---

### Author Rebuttal · Authors · 2024-08-06

# Global Response to AC and all reviewers
Thanks to all reviewers for careful reading and thoughtful feedback. We are grateful for acknowledging our method "looks promising as a general technique for sEEG data" and that our dataset “is likely to become the basis of benchmarks for future research in this area” (Reviewer gwJp). We are also excited that our work "presents strong results of high significance for the field of speech decoding" (Reviewer 1qa8). Further, we are delighted to hear our method "is rather novel and sheds light on the BCI community" (Reviewer bbQK).

In response to reviewers' comments, we performed additional experiments: 8 new model comparison experiments and 5 new ablation experiments, summarizing the additional studies. Please see the general rebuttal pdf for details. We will add these new experiments to our revised paper.

## About Model Comparison Experiment
To support the effectiveness of Du-IN for speech decoding, we added 1 supervised baseline (DeWave-CLS), 6 self-supervised baselines (TS-TCC, Neuro-BERT, original BrainBERT, original Brant, LaBraM, LaBraM-PopT) and 1 Du-IN variant (Du-IN (poms), pre-trained on multi-subjects), as a solution to similar concerns raised by some reviewers. These concerns include:
 - Insufficient comparison experiments (Reviewer t1MV, eGF6).
 - Not explicitly modeling multiple subjects (Reviewer eGF6, bbQK).
 - LaBraM comparison (Reviewer gwJp).

All the results can be found in Table 1 in the rebuttal pdf. All codes and checkpoints are available in the Anonymous Github Repo at line 328 in the initial manuscript.  First of all, we want to emphasize that **all baselines are reproduced strictly following to their original code**. Besides, all experiments are conducted on the same machine with the same set of random seeds, and **the train/valid/test splits are the same across different models**.
 - **Our model comparisons are indeed fair and comprehensive.**

For BrainBERT and Brant, we provide their original code and checkpoints, and you can directly load their weights using our model to check whether reproductions are the same. The reported performance of BrainBERT(7.5%) and Brant(12.42%) in the initial manuscript is after hyperparameter optimization. The performance of original BrainBERT(5.65%) and Brant(8.79%) is even lower. LaBraM(11.53%) and LaBraM-PopT(11.78%, which replace learnable spatial embeddings with hard-coded ones proposed by [R1] to enable pre-training on multiple subjects) perform similarly to them. Their relatively low performances are mainly because they don’t encourage region-level embedding either from architecture[R2] or loss[R1], which makes them hard to decode speech. MMM[R2] introduces an information bottleneck to encourage integrating channel-level embeddings into region-level embeddings. PopT[R1] does this due to their pre-training task. Our model directly uses depthwise convolution to get such region-level embeddings, that is why our model outperforms them by a 50% margin.

Additionally, we add TS-TCC(24.85%), Neuro-BERT(49.51%), and DeWave-CLS(32.43%) to further address the concerns about insufficient comparison experiments. Our Du-IN model (62.70%) still performs the highest among them.

We further provide Du-IN (poms) (59.18%), which is pre-trained on all 12 subjects and provides a fair comparison with BrainBERT and Brant, to address the concerns about modeling multi-subjects.

## About ablation experiment
To further demonstrate why our proposed model works for sEEG speech decoding, we conduct additional ablation experiments on our model, as a solution to similar concerns raised by some reviewers. All the results can be found in the rebuttal pdf:
 - Ablations on effectiveness of region-specific channel selection(Table 2).
 - Ablations on impact of mask ratios(Table 3).
 - Ablations on effectiveness of vector-quantized neural signal prediction(Table 4).
 - Ablations on impact of the pre-training epochs (of the Du-IN VQ-VAE and MAE)(Table 5,6).

The first ablations demonstrate that sEEG signals are fundamentally different from EEG signals due to the high information density and the high specificity of different regions. Other ablations demonstrate the robustness of our model.

## About specific responses
We have individually addressed all of your comments below, specifically addressing each reviewer’s concerns in the corresponding responses. Please note that in our responses, references in the format “[R1]” indicate citations that are newly added in the rebuttal, while references in the format "[1]" are citations from the original manuscript.

We have dedicated significant effort to improving our manuscript, and we sincerely hope that our responses will be informative and valuable. We would love to receive your further feedback.

**Reference:**

[R1]Chau G, Wang C, Talukder S, et al. Population Transformer: Learning Population-level Representations of Intracranial Activity[J]. ArXiv, 2024.

[R2]Yi K, Wang Y, Ren K, et al. Learning topology-agnostic eeg representations with geometry-aware modeling[J]. Advances in Neural Information Processing Systems, 2024, 36.

---

### Decision · Program_Chairs · 2024-09-25

**Decision:**

Accept (poster)

**Comment:**

This work has received feedback from 5 reviewers two of which are championing the paper (revs 1qa8 and gwJp). Novelty of the contribution is challenged by certain reviewers that argue that the work has strong similarities with LaBraM and is strongly inspired by BeiT. Yet the present contribution is here applied to sEEG and not EEG or computer vision, and results suggest that Du-IN is a novel method offering a strong performance on a rich decoding task (yet not on open vocabulary). Besides code is provided at https://anonymous.4open.science/r/Du-IN-3B8F/README.md rather cleanly, and authors promise to release the data upon acceptance which should enable progress on this task by the ML community.

For these reason I recommend this work for acceptance.

Yet, I am asking the authors to clearly acknowledge what figures are paragraphs are taken or inspired from the Labram paper (https://arxiv.org/pdf/2405.18765) (cf fig1, fig2, "transformer encoder" paragraph) in the camera ready version.